# Genetic analysis of F1 cluster phages that infect *Mycobacterium smegmatis* identifies two distinct holin-like proteins that regulate the host lysis event

**Richard S. Pollenz**[1]*, **Kira Ruiz-Houston**[2], **Wynter Dean**[1], **Loc Nguyen**[1]

**1** Department of Molecular Biosciences, University of South Florida, Tampa, Florida, United States of America, **2** Department of Microbiology, University of Wisconsin, Madison, Wisconsin, United States of America

* pollenz@usf.edu

## Abstract

Phages Girr and NormanBulbieJr (NBJ) infect Gram-positive *Mycobacterium smegmatis* mc² 155. Both phages contain conserved lysis cassettes that harbor two endolysin genes (*lysin A* and *lysin B*) and two genes encoding transmembrane domain (TMD) holin-like proteins. The first holin-like protein, termed LysF1a is 88 amino acids, has two TMDs and a predicted N-in-C-in membrane topology. The second, termed LysF1b, has a single N-terminal TMD and a predicted N-out-C-in topology making it distinct from the type III holins or spanins in size and membrane topology. Deletion of *lysF1b* results in severe lysis defect phenotypes manifest by reduced plaque size and changes to lysis timing in liquid culture. Deletion of both *lysF1a* and *lysF1b* genes results in phages that show the same lysis phenotypes as the single *lysF1b* deletion. Phages with only *lysF1b* are lysis competent and trigger lysis prematurely when exposed to energy poisons while phages with *lysF1a* or *lysF1a/lysF1b* deletions do not trigger prematurely. Deletion of genes upstream of the lysis cassette did not impact lysis phenotypes. Lysis recovery mutants were isolated from phages lacking the *lysF1b* gene and these mutants generated wild type plaque size but triggered lysis prematurely and showed ~65% reductions in burst size. Genome sequencing identified different point mutations that mapped to TMD1 or the C-terminal region of the *lysF1a* gene. Infection of an *M. smegmatis* strain that does not produce lipomannan and lipoarabinomannan by either wild type phages or phages carrying the *lysF1b* deletion showed modest plaque size increases but did not fully complement the lysis defect of phages lacking the *lysF1b* gene. Collectively, the findings show that both LysF1a and LysF1b proteins are required for efficient bacterial lysis by these F1 cluster phages. LysF1a does not function as a pure antiholin but requires the expression of the LysF1b protein for efficient lysis functioning.

**Data availability statement:** The authors affirm that all data necessary for confirming the conclusions of this article are represented fully within the article and its tables and figures. Full genome sequences of Girr and NBJ are available in GenBank through accession numbers MH669003 and MH399784, respectively. Extended data, including all original gel figures are posted to FileShare https://doi.org/10.6084/m9.figshare.31562248.

**Funding:** This project was funded by an Undergraduate Grants in Aid for Research (Functional Analysis of Novel Holin Proteins in Mycobacteria Phage D29) from USF Chapter of Sigma Xi to KRH and a USF Internal Proposal Enhancement Grant (Functional Analysis of Holin-Like Proteins from Bacteriophages that Infect Gram-Positive Bacterial Strains) to RSP. There was no additional external funding received for this study.

**Competing interests:** The authors have declared that no competing interests exist.

## Introduction

With an estimated $10^{31}$ different bacteriophage particles in the world, the pathway whereby dsDNA phages lyse their bacterial hosts after infection and replication is one of the most common cell fates on earth [1]. Extensive biochemical and genetic work with dsDNA phages that infect Gram-negative hosts has established a general three-step model for the triggering of the lysis pathway that involves three distinct sets of proteins. The first are endolysin/s that have defined catalytic activity to lyse the peptidoglycan (PG) layer [2]. The second are holin or antiholin proteins that have between 1–4 transmembrane domain helices (TMD). These proteins work together to, 1) control the timing of the lysis event following phage infection [3,4], 2) assist with the destabilization of the membrane proton motive force (PMF) [3,4] and 3) in the canonical holin models, generate large 1μm "pores" in the inner membrane (IM) to allow the exit of the endolysin enzymes to the periplasmic space where they can access and cleave the PG layer [5,6]. Finally, there are secreted membrane proteins and lipoproteins termed spanins that are associated with the IM and outer membrane (OM). There proteins interact after the PG layer is cleaved to generates the lesion across the IM and OM that allows for the efficient release of the phage progeny [7–9].

dsDNA phages such as lambda, P2, and phage 21 all have genomes < 50,00 bp and have a clustered lysis cassette that contain all the genes necessary to mediate the bacterial lysis event [10–12]. Phage lambda encodes the canonical type I holin protein that has 3TMDs and forms large oligomers that create 1μm holes in the IM to allow exit of the endolysin from the cytoplasm to the periplasmic space [12]. Phage 21 defines the type II pinholin lysis pathway and utilizes a 2TMD holin protein that forms ~1,000 heptamers in the IM. The pinholins create only nanometer size holes that are not big enough to allow exit of the endolysin, and they function to disrupt the PMF and trigger lysis. Thus, phages that encode pinholins also encode endolysins that carry secretion-arrest-release (SAR) signals and the SAR-endolysin is held in an inactive state in the IM prior to triggering. The disruption of the PMF causes the release of the IM tethered SAR-endolysin to the periplasmic space where it cleaves the PG [4,13,14]. In both lambda and phage 21, a single open reading frame with two in-frame start codons (dual start motif) encodes two holin proteins that only differ by the addition of 1–2 N-terminal amino acids [10,11,14,15]. The longer version of each holin functions as the inactive antiholin and in both cases a percentage of the holin forms dimers with the antiholin that holds that holin in an inactive state prior to triggering. Upon triggering, the antiholin assumes active holin function greatly increasing the amount of active holin protein available for hole formation. In phages with large genomes, such as P1 and T4, the holin, antiholin, endolysin, spanin and lysis inhibiter (LIN) rI and rIII genes can be bioinformatically characterized throughout the genome and are not in a defined cassette [10,11,16,17]. The type III holin from phage T4 has a single N-terminal TMD with an N-in-C-out topology with the bulk of the sequence oriented in the periplasmic space [16,17]. T4 phage encodes two different antiholins and LIN proteins that modulate the holin lysis timing and participate in the lysis inhibition activity of phage T4 [16,17]. Thus, a common theme in the phage-mediated lysis pathway of lambda, P1, P2 and T4 is the utilization of holins and antiholins to

regulate the lysis event. Phage T7 on the other hand, has only 1 annotated holin and no identified antiholin and deletion of the holin is not lethal but results in various lysis phenotypes suggesting that other unidentified proteins may also participate in the lysis pathway [18]. Another variation of the lysis pathway has recently been shown for the Enterococcal phage Mu [19]. Mu does not utilize canonical holins and antiholins in the bacterial lysis pathway and instead uses a novel 1TMD protein termed a releasin protein to release the SAR-endolysin from the IM so it can cleave the PG [19]. As with lambda and phage 21, Mu contains canonical spanins to create the final lesion for progeny release once the PG has been cleaved. It is speculated that Mu does not utilize the holin/antiholin regulated lysis mechanism because it copies its genome via replicative transposition during the lytic cycle and has less selective pressure due to transposition immunity [19]. This pathway adds yet another variation in phage-mediated bacterial lysis and it is striking that all the E. coli phages that have been rigorously studied utilize a similar set of tools to complete the lysis pathway through distinct molecular mechanisms.

There are 1000s of annotated dsDNA phages that infect Gram-positive hosts, yet the molecular mechanism that these pages use to lyse their hosts is not resolved to the level of the E. coli phages. Although it is hypothesized that the lysis mechanism in these phages will utilize a similar set of proteins as described for phages that infect Gram-negative bacteria, the bulk of biochemical and genetic work has focused more on the enzymes that target the cell wall than on the holins, possibly because these enzymes may be employed as antibiotic agents [20–22]. Gram-positive bacteria lack a formal OM and instead have a much more complex PG layer and an extended structural cell wall components beyond the PG such as arabinogalactans and mycolic acids [23–26]. Thus, addition of an endolysin to Gram-positive bacteria can result in cell death. Bioinformatic, biochemical and genetic analysis has identified several types of phage-encoded endolysins that target different regions of the cell wall. The first type of endolysin is termed lysin A and targets distinct domains of the peptidoglycan [21,27]. These enzymes are highly modular, do not contain canonical SAR domains, and deletion studies show that the lysin A is essential to phage propagation [28,29]. It is unknown if the lysin A is released to the peptidoglycan layer through holins since some studies show that exogenous expression of selected lysin A enzymes can be cytotoxic to M. smegmatis [21,30–33]. Many phages that infect Gram-positive hosts also contain a second endolysin termed lysin B that targets the linkages in the mycolic acid layer of the cell wall [28,31,34,35]. It has been hypothesized that lysin B provides a fitness advantage for optimal bacterial lysis and phage release since deletion of lysin B results in viable phages with reduced plaque and burst size [28].

In phages that infect Gram-positive hosts the presence of genes that encode holin-like proteins has primarily been characterized by bioinformatic analysis and many of the phages contain more than one gene encoding a TMD protein that could serve as a holin or antiholin. Currently there are reports on host lysis by phages that infect Staphylococcus aureus [36–38], Bacillus sp. [39,40], Lactococcal sp. [41], Oenococcus oeni [42,43], Streptococcus sp [44], Gordonia rubripertincta [45–47] and Mycobacterium sp. [48–55]. The holin function for the encoded TMD proteins in most of these reports is based primarily on the proximity of the TMD gene/s near an endolysin. Where biochemical work is reported, the majority has been carried out using exogenous expression of the TMD genes in E. coli. However, there are several reports on phages that infect Mycobacterium smegmatis where gene deletion has been utilized to evaluate lysis function of the predicted holin proteins.

The first study evaluated two genes encoding TMD proteins that are found directly downstream of the lysin A and lysin B in the F1 cluster phage Ms6 [50–52]. Ms6 gene 26 encodes an 88 amino acid protein with 2TMDs while gene 27 encodes a 124 amino acid protein with 1TMD. Deletion of gene 26 resulted in reduced lysis timing, reduced plaque size but wild type burst size, while deletion of gene 27 resulted in delayed lysis timing, increased plaque size and increased burst size. However, there are limitations to these findings because holin function was not verified using energy poisons and the mutant phages were not tested in liquid lysis assays in the M. smegmatis host versus E. coli. In addition, the experiments to assess function and interaction of the two proteins were carried out using recombinant E. coli expression systems that are not physiologic. Thus, while it appears that both proteins are needed for the lysis event, it is unclear how each is contributing to the lysis of the M. smegmatis host.

The second set of studies evaluated phage D29, a cluster A2 phage that also infects *M. smegmatis* [53–56]. D29 has a lysis cassette with a *lysin A* (gene *10*), 2TMD holin (gene *11*) and a *lysin B* (gene *12*). It is important to note that the 2TMD holin of D29 is not an ortholog to the 2TMD protein in Ms6 and differs significantly in amino acid sequence and size. The lysis cassette that is distinct from most other actinobacteriophages in that it has no other genes encoding holin-like TMD proteins within or near the putative lysis cassette. Deletion of gene *11* results in a viable phage with a different phenotype that the 2TMD deletion in Ms6 [53]. D29_*Δ11* phage shows a modest 30% reduced plaque size, slightly delayed lysis timing and reduced burst size [53]. The lack of a more severe lysis phenotype is surprising given that gp11 is the only defined holin, especially since the D29 lysin A does not have a defined signal peptide or SAR-like domain for export [32,33]. However, it has been reported that the D29 lysin A may be exported prior to lysis in a yet to be defined pathway that appears to be independent of the holin [32,33]. Alternatively, the modest lysis phenotype could also indicate that D29 has additional holin-like genes that are not located within the defined "lysis cassette" region. Such a scenario has recently been described in a bioinformatic study of the lysis cassettes of 77 phages that infect the Gram-positive *Gordonia rubripertincta* strain [46]. This study showed that there was a high level of diversity of the lysis cassette gene organization and that putative holin-like genes could be identified in genomic regions distal to the endolysin containing lysis cassettes [46]. This study also showed that most of the phages evaluated had at least two holin-like genes.

In summary, there is limited biochemical and genetic analysis of host lysis pathways by phages that infect Gram-positive hosts, especially within the host bacteria. One of the most intriguing findings from the annotation of phages that infect Gram-positive hosts is the presence of multiple genes that encode predicted holin-like TMD proteins in the defined lysis cassettes. Since antiholins have not been experimentally identified in phages that infect Gram-positive hosts, one interpretation of these data are that holins and antiholins are expressed from distinct genes as shown for *E. coli* phages P2 and T4 [16,17,57,58]. It is also possible that some of the TMD genes encode proteins with spanin-like functions that interact with the complex cell wall of the Gram-positive bacteria. This hypothesis is supported by a report that identified a 177 amino acid protein termed LysZ, that is encoded by phages that infect *Corynebacterium glutamicum* and may function to disrupt the integrity of the bacterial cell wall [59]. The current study aims to evaluate how two predicted holin-like TMD proteins encoded by phages Girr and NormanBulbieJr (NBJ) function in the lysis pathway of *Mycobacterium smegmatis* mc$^2$155 by utilizing recombineered phages and multiple lysis assays under physiological growth conditions.

## Materials and methods

### Bioinformatic analysis of transmembrane domains and AlphaFold3 analysis

The identification of transmembrane domains (TMD) and protein topologies was carried out essentially as described by Pollenz et al., 2024 [46] utilizing Deep TMHMM (v1.0.24) [60], TOPCONS (v2.0) [61], SignalP-5.0 and SignalP-6.0 [62], LipoP 1.0 [61] and HHpred (databases: PDB_mmCIF70_30_Mar; Pfam-A_v37) [63]. All membrane proteins described in this report had 100% consensus predictions of TMD domain location, number and protein topology in all programs. Structural predictions of proteins and protein-protein interactions were determined using AlphFold3 [64]. For single proteins, pTM scores >50% indicate high confidence in the model predictions. For protein-protein interactions, ipTM scores <90% are considered low confidence and should be used with caution [65]. All amino acid alignments were performed using Clustal Omega [66].

### Bacterial strains, growth and plaque assay

The list of bacterial strains, phage and plasmids utilized in this study is presented in Table 1. *Mycobacterium smegmatis* mc$^2$155 [67] was used as wild type host for all phage related experiments. *M. smegmatis* ΔmtpA and *M. smegmatis* ΔmtpA-comp are described in [68]. *M. smegmatis* strains were propagated on Middlebrook 7H10 agar supplemented with 10% Albumen Dextrose Complex (ADC; 5% albumen, 2% dextrose, 154mM NaCl) at 37°C. To generate saturated stock

**Table 1. Bacterial strains, phages and plasmids.**

| Bacterial Strains | Genotype/Features | Reference/Source |
|---|---|---|
| *M. smegmatis* | *Mycobacterium smegmatis* mc² 155 | 67 |
| *M. smegmatis ΔmptA* | *Mycobacterium smegmatis* mc² 155 lacking the mptA enzyme | 68 |
| *M. smegmatis ΔmptA-comp* | *ΔmptA* strain carrying complementing mtpA expression plasmid | 68 |
| *E. coli* | *Escherichia coli* NEB5α F'I^Q | New England Biolabs |

| Bacteriophage | Genotype/Features | Source |
|---|---|---|
| Phage Girr_WT | Mycobacteriophage Girr | 70. PhagesDB: https://phagesdb.org/phages/Girr |
| Phage Girr_Δ26 | Girr_WT with deletion of gene *26* | This study |
| Phage Girr_Δ29–20 | Girr_WT with deletion of gene *29* clone 20 | This study |
| Phage Girr_Δ29–25 | Girr_WT with deletion of gene *29* clone 25 | This study |
| Phage Girr_Δ27–30 | Girr_WT with deletion of genes 27–30 | This study |
| Phage Girr_ΔlysF1a | Girr_WT with deletion of gene *34* | This study |
| Phage Girr_ΔlysF1b | Girr_WT with deletion of gene *35* | This study |
| Phage Girr_ΔlysF1a/_ΔlysF1b | Girr_WT with deletion of gene *34* and gene *35* | This study |
| Phage NBJ_WT | Mycobacteriophage NormanBulbieJr | 71. PhagesDB: https://phagesdb.org/phages/NormanBulbieJr |
| Phage NBJ_Δ29 | NBJ_WT with deletion of gene *29* | This study |
| Phage NBJ_ΔlysF1a | NBJ_WT with deletion of gene *33* | 71 |
| Phage NBJ_ΔlysF1b | NBJ_WT with deletion of gene *34* | This study |
| Phage NBJ_ΔlysF1a/_ΔlysF1b | NBJ_WT with deletion of gene 32 and gene *33* | This study |
| Phage D29_ | Mycobacteriophage D29 | 56 |

| Plasmids | Features | Source |
|---|---|---|
| pExTra-Girr_*29* | pExTra01::(Ptet-Girr_*34*-mcherry); Kan^R | 70 |
| pExTra-Girr_*34* | pExTra01::(Ptet-Girr_*34*-mcherry); Kan^R | 70 |
| pExTra-Girr_*35* | pExTra01::(Ptet-Girr_*35*-mcherry); Kan^R | 70 |
| pExTra-NBJ_*29* | pExTra01::(Ptet-NBJ_*29*-mcherry); Kan^R | 71 |
| pExTra-NBJ_*32* | pExTra01::(Ptet-NBJ_*32*-mcherry); Kan^R | 71 |
| pExTra-NBJ_*33* | pExTra01::(Ptet-NBJ_*33*-mcherry); Kan^R | 71 |
| pExTra-Waterfoul_*32* | pExTra01::(Ptet-Waterfoul_*32*-mcherry); Kan^R | 69 |
| pExTra-Waterfoul_*32*ΔN | pExTra01::(Ptet-Waterfoul_*32(truncation of N-terminal TMD)*-mcherry); Kan^R | This study |
| pExTra-Waterfoul_*32*ΔC-40 | pExTra0::(Ptet-Waterfoul_*32(truncation of C-terminal 40aa)*-mcherry); Kan^R | This study |
| pJV53 | Acetamide inducible plasmid with Ched9c gene *60–61*. Kan^R | 73 |

cultures, a single colony of *M. smegmatis* was propagated in 25 ml of Middlebrook 7H9 media supplemented with 10% ADC, 1 mM $CaCl_2$, 0.5% glycerol and 0.05%

Tween-80 with shaking (225 rpm) at 37°C. For all liquid lysis and one-step experiments, 100 ul of saturated stock was added to 250 ml baffled flasks containing 55 ml of supplemented 7H9 without Tween. Cultures were shaken (225 rpm)

overnight at 37°C and utilized in experiments the next day while in log phase growth ($OD_{600} < 0.8$). Titering experiments showed that an *M. smegmatis* culture at an $OD_{600}$ of 1.0 was equal to ~$6.5 \times 10^7$ CFU/ml. This value was utilized when phages were infected at different MOIs. Plaque assays were completed by mixing the appropriate amount of phage with 250ul of saturated *M. smegmatis* in Middlebrook 7H9 media supplemented with 10% ADC, 1mM $CaCl_2$ and 0.5% glycerol. The mixture was vortexed and incubated at 22°C for 10 minutes. Following the infection, the sample was mixed with 4–5ml of top agar (Middlebrook 7H9, 1mM $CaCl_2$, 0.5% agar) and plated immediately onto 7H10 agar plates. Plates were allowed to harden and placed in a 37°C bacterial incubator for 16–48 hrs. Competent *M. smegmatis* cells for recombineering or for plasmid transfection and complementation studies were prepared as described previously [29,69–71]. *Escherichia coli* NEB5α F′I^Q (New England Biolabs) was used for plasmid amplification and purification and grown in LB broth or on LB agar supplemented with 50ug/ml kanamycin sulfate and.

## Plasmid constructs, *M. smegmatis* transformation and complementation analysis

All PCR primers utilized in this study are shown in S1 Table 1 in S1 File. To generate the inducible expression plasmids pExTra_Waterfoul_*32ΔN*, pExTra_Waterfoul_*32ΔC40*, pExTra_Waterfoul_32 was PCR amplified with Q5 DNA polymerase (New England Biolabs) using Waterfoul specific primers containing with pExTra homology regions (Table S1 in S1 File). Purified PCR products (were ligated into linearized pExTra01 using isothermal assembly (NEB HiFi 2x Master Mix) as described [67–69]. Recombinant plasmids were recovered by transformation of *E. coli* NEB5α F′I^Q and colonies selected on LB agar supplemented with 50 µg/µl kanamycin sulfate. Plasmid DNA was purified using the GeneJet kit (Fisher) and all plasmids were sequence-verified by Sanger sequencing using the pExTra Seq primer (Azenta). For generating complementing strains, competent *M. smegmatis* cells were prepared as described [67–69] and 40µl mixed with 100ng plasmid DNA in a 1mm electroporation cuvette. The sample was electroporated with 1800V and the time constant set to 5.0 msec. Following electroporation, the sample was mixed with 950µl of supplemented 7H9 media without Tween and shaken (225 rpm) at 37°C for 120 minutes. Following the recovery period, 10µl-100µl aliquots were spread on 7H10 agar plates supplemented with 10 µg/µl kanamycin sulfate and incubated at 37°C for 3–5 days. To generate cultures for plaque assay, single colonies of *M. smegmatis* carrying the appropriate plasmid were harvested into 25ml of supplemented 7H9 media containing 10 µg/µl kanamycin sulfate. Cultures were shaken (225 rpm) at 37°C for 24–48 hours. Saturated cultures at $OD_{600} > 1.5$ were centrifuged at 6200 rpm for 10 minutes and the pellets resuspended in supplemented 7H9 media (without Tween) to a final $OD_{600}$ of 2.0. These cells were utilized for phage infection and then mixed with 4.5ml top agar supplemented with 10ng/µl kanamycin sulfate with or without 100ng/µl anhydrotetracycline (aTc). The mixture was added to 7H10 agar containing 10 µg/µl kanamycin sulfate supplemented with or without 100ng/µl aTc and incubated at 37°C for the times indicated in the results section.

## Bacteriophage recombineering and genome sequencing

All gblocks and PCR primers utilized in this study are shown in Table S1 in S1 File. The deletion of genes in phage Girr and NBJ were carried out using the Bacteriophage Recombineering of Electroporated DNA (BRED) procedure as described [29,72], with the following modifications. Phage genomic DNA was purified using the Wizard DNA Clean up kit (Promega). dsDNA recombineering substrates were synthesized as gblocks (Integrated DNA Technologies). To generate µg quantities of dsDNA substrates, gblocks were solubilized in water to 10ng/µl and PCR amplified with gblock specific primers using 2X Q5 master mix (New England Biolabs). PCR products were purified using Zymoclean DNA recovery (Zymoresearch). To generate recombinant phage, 200−400ng dsDNA substrate was mixed with 100ng phage genomic DNA and incubated on ice for 10min. The DNA mixture was added to 100µl of recombineering *M. smegmatis* in a 2mm electroporation cuvette and electroporated at 2500V with the time constant set to 20.0 msec. Following electroporation, 900µl of supplemented 7H9 media was added to the cells and they were shaken (225 rpm) at 37°C for 90 minutes.

Following the recovery period, the sample was mixed with 250μl of saturated *M. smegmatis* and plated with 5 ml 7H9 top agar onto 7H10 agar plates. Plates were incubated at 37°C for 18−24 hours to recover primary plaques. Primary plaques were picked into 120μl of phage buffer (10mM Tris-HCl, pH 7.5; 10mM MgSO$_4$, 68.5mM NaCl; 1mM CaCl$_2$), vortexed, and incubated at 22°C for 2−8 hrs. The presence of mutant phage genomes was determined by Deletion Amplification Detection Assay (DADA) PCR [29] using a 20μl reaction with: 1μl of plaque sample, 2μl of 0.2 μM DADA primer mix (1:1 mix of For and Rev primers), 5μl glycerol (40%), 2μl water and 10μl OneTaq 2x Master Mix (New England Biolabs). PCR conditions were: 94°C for 1 minute followed by 35 cycles of 94°C-1min/66°C 1 min and a final 68°C step for 5 minutes. Positive primary plaque samples were serially diluted and plated on wild type or complementing *M. smegmatis* as indicated. Secondary plaques were picked as above and screened by standard PCR using gene specific flanking primers. Pure populations of mutant phage were evaluated by full genome sequencing as described [73,74]. Briefly, Genomic DNA was used to create sequencing libraries with the NEB XLEP-P1 kit. Sequencing was performed by the Pittsburgh Bacteriophage Institute and the library run on an Illumina NextSeq 1000 instrument. Raw reads were trimmed with cutadapt 4.7 (using the option: –nextseq-trim 30) and filtered with skewer 0.2.2 (using the options: -q 20 -Q 30 -n -l 50) prior to assembly. The resulting sequences were assembled using Consed (v29.0) with Unicycler (v5.0) and contigs checked for completeness, accuracy, and genome termini. Final phage contigs were aligned to wild type phage to identify deletions and point mutations.

## Plaque size analysis

Plaques were generated using the plaque assay procedure described above. For experiments where plaque size would be evaluated between several different phages, identical bacterial samples, incubation times, phage concentrations and top agar amounts were utilized. Plaque size was evaluated using ImageJ software [75] and by manual measurement using photos embedded in PowerPoint and custom rulers. In most experiments, results are presented for an average of 50 plaques and the statistical significance of measurements determined by unpaired t-test (https://www.graphpad.com/quickcalcs/ttest1/). Plaque volume was calculated by the formula $\pi r^2 h$ where r is the radius of the plaque and h was set to 1 mm for the top agar thickness.

## Liquid lysis assay and cell viability assay from liquid cultures

Liquid lysis assays were carried out essentially as described by Payne et al., 2009 [28] with the following modifications. Briefly, overnight cultures of *M. smegmatis* in log phase growth were diluted to an OD$_{600}$ of 0.25–0.30 in supplemented 7H9 without Tween and 40 ml of culture was added to 250 ml baffled flasks. High titer phage lysate was added to the cultures to create an MOI of 10 and the mixture set at 37°C for 30 minutes without shaking. T$_0$ for all experiments was the time of phage addition. After 30 minutes (T$_{30}$), the cultures were constantly shaken a 225 rpm and aliquots of cells removed with a syringe and sterile needle from the shaking culture without stopping the incubator. The OD$_{600}$ of the sample was immediately recorded for each aliquot using a GENESYS spectrophotometer (Fisher Scientific). In some experiments, KCN (1M in water), or CHCl$_3$ (100%) was added directly to the shaking culture to produce a final concentration of 10mM or 1%, respectively. All liquid lysis experiments were repeated at least five separate times and representative experiments are presented.

To assess if there were viable cells surviving in the phage infected cultures, bacteria was aliquoted from a single stock sample at an OD$_{600}$ of 0.25–0.27 and infected at an MOI of 10 as described above. An aliquot of culture was removed from uninfected and infected cultures at the times specified in the experiment, evaluated at OD$_{600}$ and serial diluted in 7H9 neat. 250μl aliquots of the dilutions were spread on 7H10 agar plates and incubated at 37°C until colonies could be counted (4–5 days). To assure that surviving colonies all arose from infected cells and were not the result of phage resistant *M. smegmatis* or contamination, lytic phage D29 [56] was used as a positive control with an expected outcome of zero viable cells.

## ATP assay

The presence of ATP in culture supernatants was determined utilizing the ENLIGHTEN kit (Promega). Briefly, liquid lysis assays were carried out as described above. A 500ul aliquot of culture was removed at indicated time points and immediately passed through a 0.2μ syringe filter to remove bacteria. The sample was frozen at -20°C prior to analysis. The presence of ATP in culture supernatants was determined by mixing 5μl with 50μl ENLIGHTEN reagent (Promega), mixing for 5 seconds and then reading the integrated luminescence over 10 seconds using a GloMax luminometer (Promega). To assure that the readings were within the linear range of the instrument an ATP standard curve was utilized.

## One-step growth curves and burst size determination

Several experimental procedures have been utilized for one-step growth curves with Gram-positive bacteria [33,49,51,53]. In our hands, none of these procedures generated consistent results or provided the ability to directly measure burst size. Thus, we developed a one-step protocol that is illustrated in S2 Figure 1 in S1 File and described below. To prepare the cells, the $OD_{600}$ of fresh overnight cultures of *M. smegmatis* in log growth were determined and 10ml centrifuged at 3,000rpm for 10 minutes. The cells were resuspended in supplemented 7H9 to a final concentration equal to an $OD_{600}$ of 3.2/ml. 250μl of cells (~5 x $10^7$ cfu) were added to microfuge tubes and ~5 x $10^7$ phage added to the cells to create an MOI of 1. The same amount of phage was also added to 250μl of supplemented 7H9 media without cells to verify the titer and determine the level of phage adsorption. The mixture was vortexed and set at 37°C for 30 minutes without shaking. $T_0$ for all experiments was the time of phage addition to the cells. After 30 minutes, 750μl of supplemented 7H9 was added to all samples. Samples were vortexed and centrifuged at 6,200rpm for 4 minutes to separate unabsorbed phage. The supernatant was saved for analysis of titer and adsorption, and the cell pellet immediately resuspended in 1ml supplemented 7H9, vortexed and diluted 1:10,000 into 70ml of supplemented 7H9 in baffled 250ml flasks. An aliquot was removed and 5μl, 12.5μl and 25μl used for plaque assay to determine the number of infectious centers to determine the precise number of infected cells. A second aliquot was removed and filtered through a 0.2μ syringe filter and 150μl utilized for plaque assay to test for unabsorbed (free) phage at $T_{45}$ minutes (the first one-step data point). The flask was placed in a 37°C incubator and constantly shaken at 225rpm for the duration of the experiment. At specified time points, 500μl aliquots were removed from the shaking flasks and each sample immediately passed through a 0.2μ syringe filter to remove cells and collect free phage. The amount of phage in each sample was determined by performing plaque assays using 150μl of serially diluted samples (usually $10^0$ to $10^{-4}$ dilutions were sufficient). The phage titer in the flask, expressed as PFU/ml was determined at each time point by counting the plaques on each plate, dividing by the volume of phage used in the infection (150μl), multiplying by the dilution factor and then multiplying by $10^3$. The number of infectious centers in the infected 70ml culture was determined by counting the plaques, dividing by the volume used for the plaque assay and multiplying by 70,000. The original phage titer and the percentage of adsorbed phage was determined by spot titers and plaque assay of the serially diluted supernatants collected from the samples of phage with and without the added cells. Since the number of cells and phage used in these experiments was held constant, this procedure consistently produced ~3.5–5.5 x $10^7$ infectious centers. This procedure also resulted in essentially zero unabsorbed phage in the culture at the start of the one-step assay. Burst size at all time points could be directly determined by dividing the total phage in the 70ml culture by the number of infectious centers. All phages were evaluated through a minimum of three separate assays. A representative data set showing the raw data for adsorption, infections centers and a one-step assay is provided in S3 Figure 2 and S4 Figure 3 in S2 File.

## Results and discussion

### Topologies of holin-like TMD proteins within the lysis cassettes of F1 cluster phages Girr and NormanBulbieJr

To investigate the contribution of predicted holin-like proteins in the lysis of *M. smegmatis*, it was desirable to begin work in phages that had a defined lysis cassette and prior wet-lab data. F1 cluster phages Girr and NormanBulbieJr (NBJ) were

selected since both phages have a define lysis cassette and a full genome cytotoxicity screen has been reported for both phages [70,71]. Girr and NBJ have an average nucleotide identity of 89.86% but only share 69 of 102 genes. Both phages have a defined lysis cassette organization located downstream of the *tape measure* and *minor tail* genes and encode a Lysin A enzyme predicted to target the peptidoglycan layer and a Lysin B that targets the linkage of the arabinogalactins to the mycolic acids (Fig 1A). The *lysin B* gene is directly followed by two genes encoding proteins that have predicted TMDs (Girr *34* and *35* and NBJ *32* and *33*).

It was first important to bioinformatically evaluate the predicted topologies of these TMD proteins and determine how they compared to previously characterized holins or other phage proteins involved in bacterial lysis. Girr *34* and NBJ *32* encode small 80aa and 77aa proteins, respectively, that have two predicted TMD regions and an N-in-C-in topology by all TMD prediction programs (Fig 1B). The proteins are 97% identical and have cytoplasmic N and C-terminal domains that each contain four charged amino acids. The presence of two TMDs and the N-in-C-in topology is like the phage 21 $S^{21}$ type II pinholin (Fig 1B and 1C). However, the presence of a pinholin is typically associated with a SAR-endolysin that is secreted to the periplasmic space prior to lysis triggering [10,11,13]. Analysis of the Girr or NBJ Lysin A proteins by the various TMD prediction programs and AlpahFold3 (S5 Figure 4 in S2 File), show no evidence of SAR domains or signal sequences and this is consistent with previous analysis of Lysin A proteins from phages that infect *M. smegmatis* [21]. Thus, it is critical to contrast several structural differences of gp32 and gp34 to the canonical $S^{21}68$ pinholin and $S^{21}71$ antiholin.

First, the $S^{21}68$ holin protein has only 3 N-terminal amino acids prior to the TMD1 and this feature is a critical to its function as a holin and allows the TMD1 to become cytoplasmic (Fig 1B and 1C) [11–15]. Second, the $S^{21}$ gene has a dual start motif that produces both the $S^{21}68$ holin protein and the $S^{21}71$ antiholin after phage infection [11–15]. The difference in the two proteins is the addition of the extra basic K residue to the N-terminal cytoplasmic domain of $S^{21}71$ that limits the TMD1 from exiting the membrane to the same degree as the $S^{21}68$ holin (Fig 1B and 1C; [11–15]). Thus, the expression of $S^{21}71$ keeps a percentage of the $S^{21}68$ in an inactive dimer until triggering at which time the $S^{21}71$ converts to an active holin [13–15,76,77]. Third, the $S^{21}68_{IRS}$ protein is an engineered version of $S^{21}68$ that adds the MRYIRS motif and 4 charged residues in the N-terminal cytoplasmic domain that creates a dominant negative version of the antiholin that is poorly triggered to an active holin [76]. Neither Girr gene *34* nor NBJ gene *32* have a dual start motif that generates multiple products of the same open reading frame. Genes *34* and *32* encode a single protein product that has a predicted N-terminal cytoplasmic domain of 13–16 amino acids of which four are charged and two others are polar. Therefore, the N-terminal topology and the number of charged amino acids in Girr gp34 and NBJ gp32 are more similar with the $S^{21}71$ antiholin and $S^{21}68_{IRS}$ and would predict that TMD1 of these proteins remains embedded in the membrane. In addition, the TMDs of $S^{21}$ both contain GxxxG motifs that are predicted to contribute to homotypic and/or heterotypic helix interactions of the monomers that facilitate the formation of the pinholes [76,77]. While TMD1 and TMD2 of Girr and NBJ have a high percentage of GA residues (Fig 1B), they lack defined GxxxG domains found in $S^{21}$. Finally, although AlphaFold3 does not model membrane proteins as well as soluble proteins, Girr gp34 was not modeled with high confidence into either a monomer or a heptamer compared to $S^{21}68$ (S6 Figure 5 in S2 File). Taken together, this data suggests that Girr gp34 or NBJ gp32 are unlikely to function as a type II holin. Based on the literature we will designate these proteins as LysF1a due to a predicted function in lysis (lys) and being first identified in F1 cluster phages.

Since there is a second gene encoding a protein with a single TMD directly downstream of the *lysF1a* gene, we hypothesized that this protein may also be involved in the lysis pathway. Girr *35* and NBJ *33* both have −4 bp ATGA overlaps to the upstream *lysF1a* gene and encode 98% identical 124aa proteins with a single N-terminal TMD and a predicted N-out-C-in topology by all TMD prediction programs (Fig 2A). Both proteins have a 22 amino acid extracellular domain followed by the single TMD. There are four basic residues following the TMD and AlphaFold3 predicts an extended helical cytoplasmic C-terminal domain that contains ~50% KRDE charged residues typical of holins [10,11]. Fig 3B shows the Alpha-Fold3 models and predicted topologies of representative proteins with a single TMD that are implicated in phage-mediated

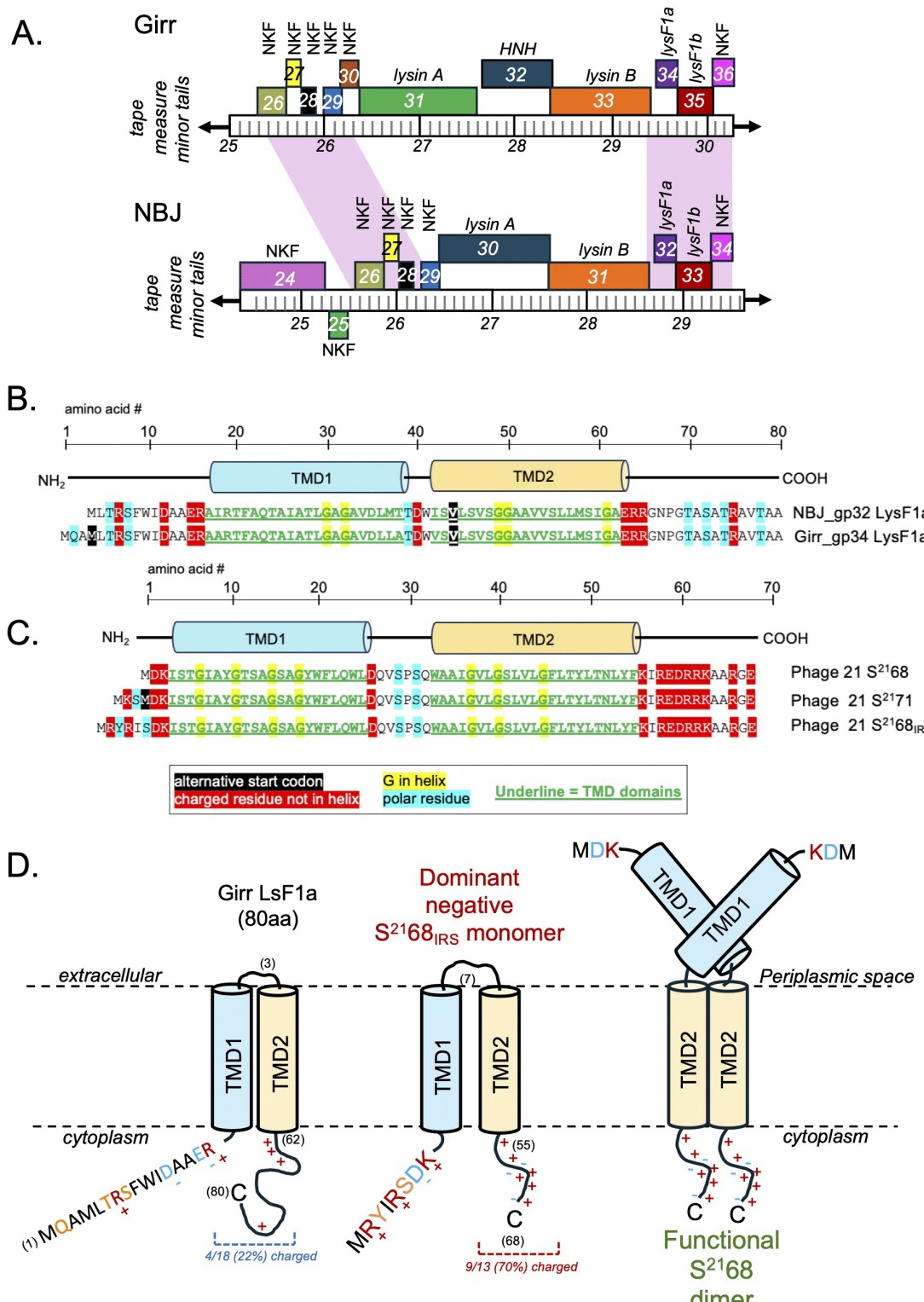

**Fig 1. Lysis cassette organization of Girr and NormanBulbieJr (NBJ) and amino acid sequence and topology of Girr and NBJ LysF1a proteins and comparison to phage 21 S²¹ pinholin. A.** Predicted protein coding genes are shown as colored boxes. The genomic location is represented by the ruler, and the numbers represent the kilobase from nucleotide 1. Both lysis cassette regions are located upstream of the *tape measure* and *minor*

*tail* genes. Genes above the genome are transcribed in the forward direct and genes below in the reverse direction. Genes with the same color encode proteins that are grouped to the same pham. Pink shading indicated nucleotide identity >90% in the shaded region. The *lysin A, lysin B,* 2TMD *lysF1a and* 1TMD *lysF1b* genes are identified. NKF = genes with no know function; *HNH* = HNH endonuclease. **B.** The amino acid sequences of Girr and NBJ LysF1a showing the location of the predicted TMDs and charged amino acids. **C.** The amino acid sequence of phage 21 $S^{21}68$ holin, $S^{21}71$ antiholin and $S^{21}68_{IRS}$ showing the location of the predicted TMDs and charged amino acids. **D.** Predicted membrane topology of Girr LysF1a, $S^{21}68_{IRS}$ and the proposed topology of the $S^{21}68$ holin homodimer with cytoplasmic TMD1.

bacterial lysis compared gp33 and gp35. Most pertinent is the type IIIa holin typified by the T protein of phage T4 that forms large 1um holes like canonical type I holins [16,17] and the recently identified type IIIb holin identified in jumbo-phage PhiKZ [78]. Neither type III holin has > 20% amino acid identity to gp33 and gp35, and both are much larger with the T having the opposite membrane topology with a globular extracellular domain critical to its holin function (Fig 2B). Although the PhiKZ holin has an extended cytoplasmic domain, the protein has a C-out-N-in topology and also has a glob-ular cytoplasmic domain. Nether have the highly charged cytoplasmic regions found in the LysF1 proteins, It is noteworthy that spanins also have a single TMD and model with an extended helical prediction, but like the T holin, these proteins are also oriented in a N-in-C-out topology that has been shown experimentally and is also predicted by the TMD prediction programs (Fig 2B; [10,11]). Interestingly, there is a recent report regarding phages that infect the Gram-positive *Coryne-bacterium glutamicum* strain. The authors have characterized a protein with 1 TMD that is termed LysZ and is proposed to interact with lipid-anchored glycans on the extracellular side of the IM to promote efficient lysis [59]. However, like the type IIIa holin, LysZ is much larger than the Girr and NBJ proteins and it is proposed to have a N-in-C-out topology that allows it to interact with the lipid-anchored glycans [59]. Thus, Girr gp35 and NBJ gp33 are most similar in structure and membrane topology to the recently identified phage Mu releasin protein (Fig 2B) [19]. Releasin however, is named due to its function to release the Mu SAR-endolysin from the membrane in a holin independent manner [19]. Mu also has < 20% amino acid identity to gp33 and gp35 and does not possess the highly charged cytoplasmic domain. Thus, of all the char-acterized proteins involved in lysis that have 1 TMD, gp35 and gp33 appear to represent a novel class of proteins and will be termed LysF1b since the gene is downstream of the *lysF1a*.

### Deletion of individual *lysF1a* and *lysF1b* genes in F1 phage are viable and results in reduced plaque size

To evaluate the function of the *lysF1a* and *lysF1b* genes, the BRED recombineering method was utilized to delete them in both Girr and NBJ as described in Methods. Several primary plaques were positive for each deletion and pure populations of Girr_Δ*lysF1a*, Girr_Δ*lysF1b*, NBJ_Δ*lysF1a*, and NBJ_Δ*lysF1b* were recovered from secondary plaques. Fig 3 shows the lysis cassette organizations of the Girr mutants and the PCR verification of the gene deletions from clonal high titer lysates. The data for NBJ is shown in S7 Figure 6 in S2 File and deletion of gene *33* is also presented in Wise et al. [71]. The genome structure of all deletions was confirmed by sequencing the immediate region of the deletion and by whole genome sequencing to validate that there were no point mutations outside of the targeted gene deletion region.

To assess the phenotypes of the mutant phage, equal concentrations of Girr_WT, Girr_Δ*lysF1a* and Girr_Δ*lysF1b* were used to infect *M. smegmatis* and the total number of recovered plaques and plaque sizes evaluated. All phages formed plaques with equivalent efficiencies under the standard assay conditions described in Methods suggesting no defects in adsorption (S8 Figure 7 in S2 File). However, both Girr_Δ*lysF1a* and Girr_Δ*lysF1b* presented with significantly smaller plaques than Girr_WT (Fig 4) as did NBJ_Δ*lysF1a* and NBJr_Δ*lysF1a* (S9 Figure 8 in S2 File). A summary of the plaque size data for both Girr and NBJ is presented in Table 2. Deletion of the *lysF1a* gene from Girr or NBJ results in ~40% reduction in plaque diameter and

a 65% reduction in plaque volume, while deletion of the *lysF1b* is more severe showing a ~65% reduction in plaque diameter and near 90% reduction in plaque volume. To validate that these plaque size reductions were directly related to the loss of *lysF1a* or *lysF1b* genes, *M. smegmatis* containing complementing expression plasmids were infected with

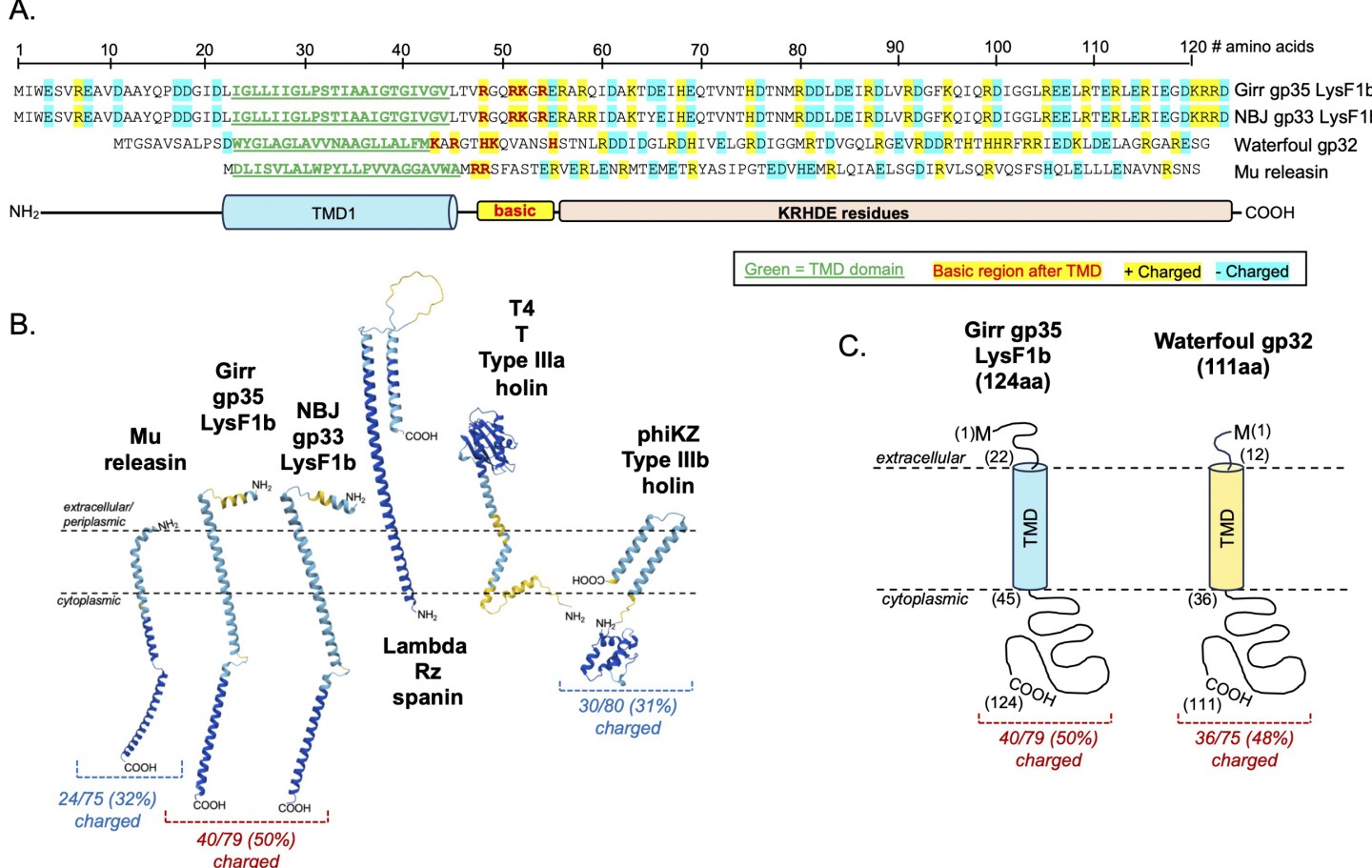

**Fig 2. Amino acid sequence and topology of lysis proteins with 1TMD. A.** The amino acid sequences of Girr LysF1b, NBJ LysF1b, Waterfoul gp32 and Mu releasin showing the location of the predicted TMD and the charged amino acids. **B.** AlphaFold3 structural predictions and membrane topologies of 1TMD proteins involved in lysis. pTM values: Mu, 0.51; NBJ LysF1b, 0.40; Girr LysF1b, 0.41; Rz, 0.57; T, 0.74; phiKZ, 0.58. **C.** Predicted topologies of Girr LysF1b and Waterfoul gp32.

the mutant phages and plaque size evaluated as described in Methods. Since both Girr LysF1b and NBJ LysF1b are highly toxic when exogenously expressed in *M. smegmatis* [70,71], complementation of the *lysF1b* deletions first utilized Waterfoul gp32. Waterfoul is a K5 cluster phage with a canonical lysis cassette and gene *32* encodes a 1TMD with the same general size and membrane topology as Girr LysF1b (Fig 2D) and is not toxic when induced by aTc in *M. smeg-matis* [67]. When M. *smegmatis* transfected with Waterfoul gp32 is infected with Girr_Δ*lysF1b* in the presence of aTc, there is a significant increase in plaque size compared to untreated cells (Fig 5A). Importantly, the increase in plaque size was not observed when the N-terminal TMD domain or the highly charged C-terminal 40 amino acids of Waterfoul gp32 were truncated (Fig 5A). Thus, the complementation by Waterfoul gp32 requires the full-length protein. Recently it has been reported that the pTet promoter in pExTra is leaky and that it is possible to complete complementation analysis in the absence of aTc [71]. Thus, *M. smegmatis* was transfected with pExTra-Girr*35* and pExTra-NBJ*33* and the resulting cells infected with Girr_Δ*lys1b* as described in Methods. Fig 5B shows that there is a significant increase in plaque size when Girr_Δ*lys1b* is plated on the complementing strains expressing the LysF1b proteins compared to cells with the base pExTra01 vector. These results support that the reduced plaque size of Girr_Δ*lysF1b* is directly attributable to the loss

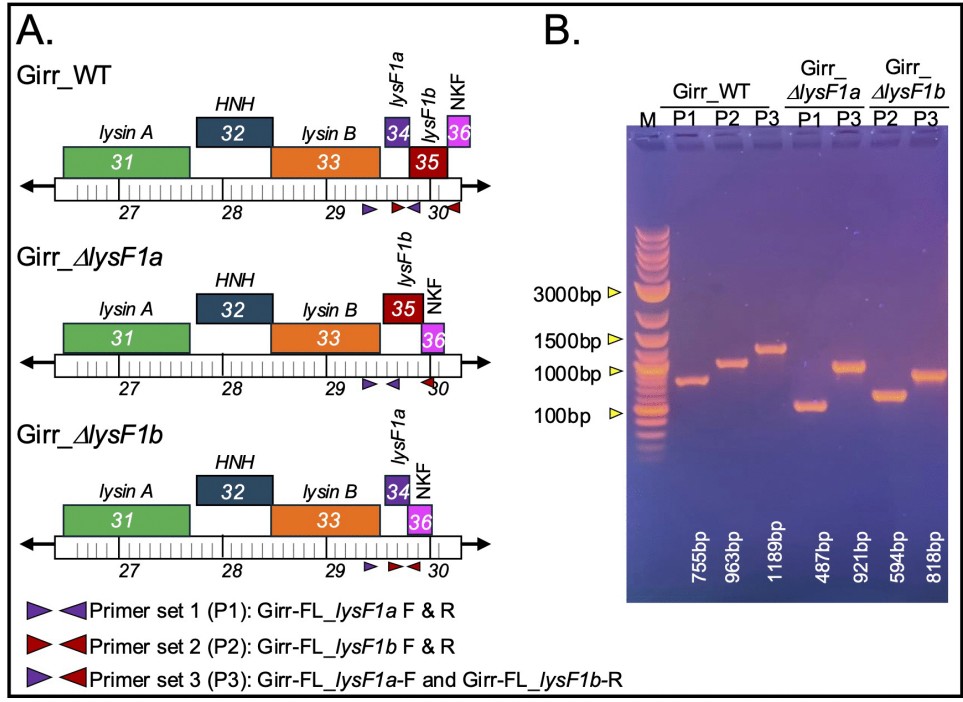

**Fig 3. Deletion of Girr *lysF1a* and *lysF1b*. A.** Genomic structure and PCR primer set locations in Girr gene deletion mutants. Primer set 1 (P1) uses primers Girr-FL_*lysF1a* (F and R) and generates a 755 bp fragment in Girr_WT and 487 bp with *lysF1a* deletion. Primer set 2 (P2) uses primers Girr-FL_*lysF1b* (F and R) and generates a 933 bp fragment in Girr_WT and 594 bp with *lysF1b* deletion. Primer set 3 (P3) uses the forward Girr-FL_*lysF1a* and reverse Girr-FL_*lysF1b* and generates a 1189 bp fragment in Girr_WT and either 921 bp with *lysF1a* deletion or 818b with *lysF1b* deletion. **B.** PCR verification of high titer lysates of Girr_WT, Girr_Δ*lysF1a* and Girr_Δ*lysF1b* using the indicated primer sets. PCR band sizes are identified and show correct deletions of the indicated genes.

of *lysF1b*. Similarly, the wild type plaque size is restored when *M. smegmatis* harboring the pExTra-Girr_*34* plasmid is infected with Girr_Δ*lysF1a* (Fig 6). Thus, the loss of either *lysF1* gene appears to create a lysis defect phenotype that is like the phenotype when the *lysin B* is deleted in phage Giles and the cell wall components remain a barrier to efficient phage release [28].

### Deletion of individual *lysF1* genes result in distinct *M. smegmatis* liquid lysis phenotypes and ATP release

A main function of holins is the control of the lysis timing by the destruction of the proton motive force (PMF) [10,11]. To assess lysis timing of Girr, NBJ and the *lysF1* mutants, liquid lysis assays were performed where the optical density ($OD_{600}$) of infected log growth *M. smegmatis* cultures is measured over time. Importantly, in all studies, once phage had been allowed to absorb for 30 minutes, cultures were continually shaken even while samples were withdrawn so that lysis would not be prematurely triggered by changing the culture conditions [11,79]. The data show that cultures infected with Girr_WT grow steadily until ~110 minutes post infection. Beginning at 110 minutes, there is a significant drop in the $OD_{600}$ of the culture (triggering event) until it reaches a baseline of $OD_{600}$ 0.10 at about 200 minutes (Fig 7A). In contrast, Girr_Δ*lysF1a* grows until ~130 minutes and then shows a reduction in $OD_{600}$ until 300 minutes that does not reach the same $OD_{600}$ baseline as Girr_WT even after 480 minutes of incubation (Fig 7A). Strikingly, cultures infected with Girr_Δ*lysF1b* do not show a drop in $OD_{600}$ but appear to cease growing after ~200 minutes with only a slight decrease in $OD_{600}$ over the remaining growth period out to 480 minutes (Fig 7A). For NBJ_WT, $OD_{600}$ begins to drop after 130 minutes and like

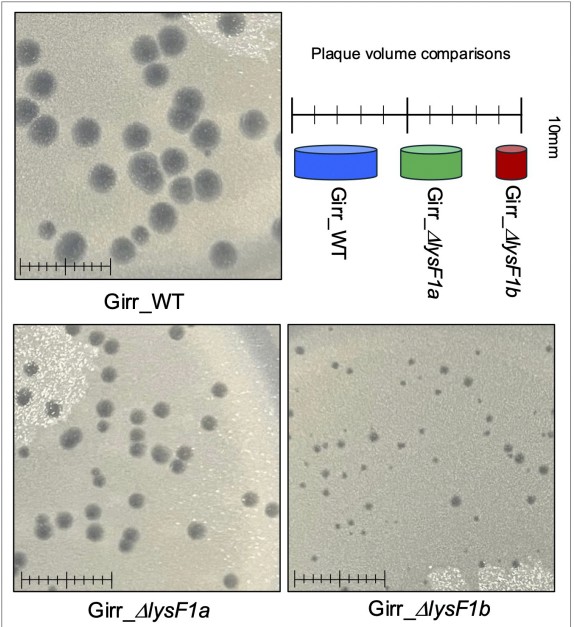

**Fig 4. Plaque sizes in Girr_WT, Girr_ΔlysF1a and Girr_ΔlysF1b.** *M. smegmatis* was infected with identical numbers of the indicated phage as detailed in Methods and plaque size determined after 36 hrs. of growth at 37°C. Scale = 1 cm. Plaque volume schematics are presented to illustrate the plaque size differences.

**Table 2. Plaque sizes in WT Girr and *lysF1a* and *lysF1b* deletion mutants.**

| PHAGE | Plaque size (mm) | % of WT (diam.) | Plaque (vol, mm³) | % of WT (plaque vol) |
|---|---|---|---|---|
| NBJ_WT | 3.49 +/- 0.533 | 100 | 11.479 | 100 |
| NBJ_ΔlysF1b | 1.23 +/- 0.225* | 35.24* | 1.425* | 12.42* |
| NBJ_ΔlysF1a | 2.05 +/- 0.660* | 58.73* | 3.96* | 34.50* |
| Girr_WT | 3.88 +/- 0.731 | 100 | 14.188 | 100 |
| Girr_ΔlysF1b | 1.28 +/- 0.376* | 32.98* | 1.544* | 10.88* |
| Girr_ΔlysF1a | 2.31 +/- 0.719* | 59.53* | 5.029* | 35.44* |

*M. smegmatis* was infected with identical numbers of phage and plaque size determined after 36 hrs. of growth at 37°C. Plaque diameter was measured using ImageJ. Data is presented as average +/- standard deviation for a minimum of 50 individual plaques. * = statistically significant from Girr or NBJ WT phage $p < 0.0001$.

Girr_WT reaches a baseline of around $OD_{600}$ 0.10 (Fig 7B). NBJ_ΔlysF1a shows a delay in lysis timing like Girr_ΔlysF1a, but the delay is more pronounced as the $OD_{600}$ does not begin to drop until ~180 minutes post infection (Fig 9B). Just like Girr_ΔlysF1b, NBJ_ΔlysF1b ceases to grow after ~200 minutes and does not show a significant reduction in $OD_{600}$ even after 480 minutes of growth (Fig 7B). Thus, in both phages, deletion of *lysF1a* results in a triggering delay while deletion of *lysF1b* results in no defined triggering event in this assay. It is interesting that Girr_WT and NBJ_WT show a ~20-minute difference in triggering time and that a difference is also observed between Girr_ΔlysF1a and NBJ_ΔlysF1a. These results may be due to differences in how the lysis cassette genes are expressed since Girr has an HNH endonuclease gene between the *lysin A* and *lysin B* (Fig 1A) and the intergenic region between the *lysin B* gene and the *lysF1a* is 8 bp in NBJ but 38 bp in Girr (S10 Figure 9 in S2 File).

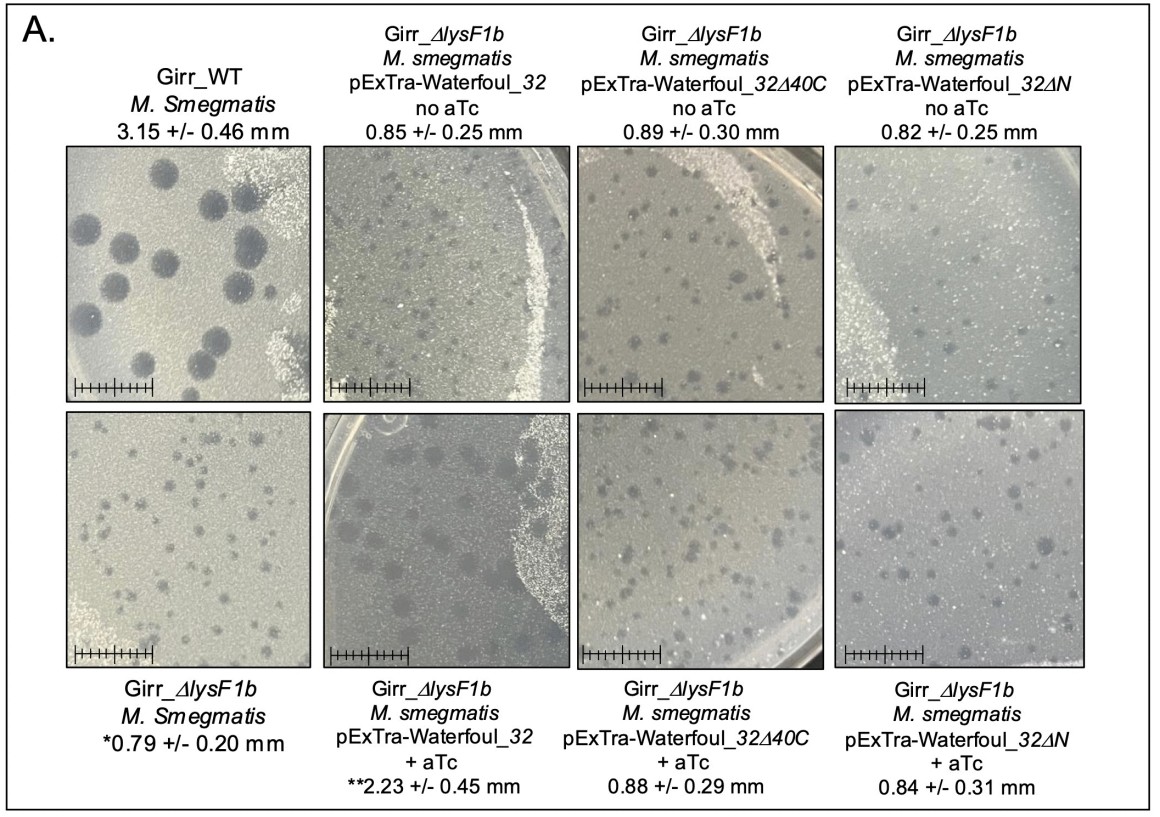

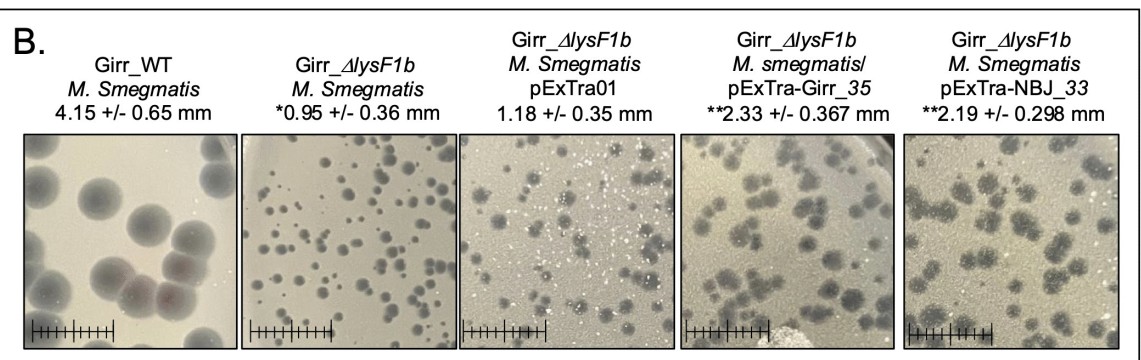

**Fig 5. Complementation of plaque phenotype in Girr_ΔlysF1b. A.** Plaque assays were completed with *M. smegmatis* using Girr_WT or Girr_ΔlysF1b as described in Methods. Plaque assays were completed with *M. smegmatis* complementing strains carrying the indicated pExTra-Waterfoul expression plasmids using Girr_ΔlysF1b as described in Methods. Plaques were evaluated following incubation at 37°C for 36 hrs. Plaques sizes were quantified using ImageJ and the average diameter +/- standard deviation for > 50 plaques is shown. * = statistically different from Girr_WT, p < 0.0001. ** = statistically different from Girr_ΔlysF1b infected on *M. smegmatis* with pExTra Waterfoul_32 with no aTc, p < 0.0001. Scale = 1 cm. **B.** Plaque assays were completed with *M. smegmatis* using Girr_WT or Girr_ΔlysF1b as described in Methods. Plaque assays were completed with *M. smegmatis* complementing strains carrying the indicated pExTra-Waterfoul expression plasmids using Girr_ΔlysF1b as described in Methods. Plaques were evaluated following incubation at 37°C for 48 hrs. Plaques sizes were quantified using ImageJ and the average diameter +/- standard deviation for > 50 plaques is shown * = statistically different from Girr_WT, *p < 0.0001*. ** = statistically different from Girr_ΔlysF1b infected on *M. smegmatis* with pExTra01, *p < 0.0001*. Scale = 1 cm.

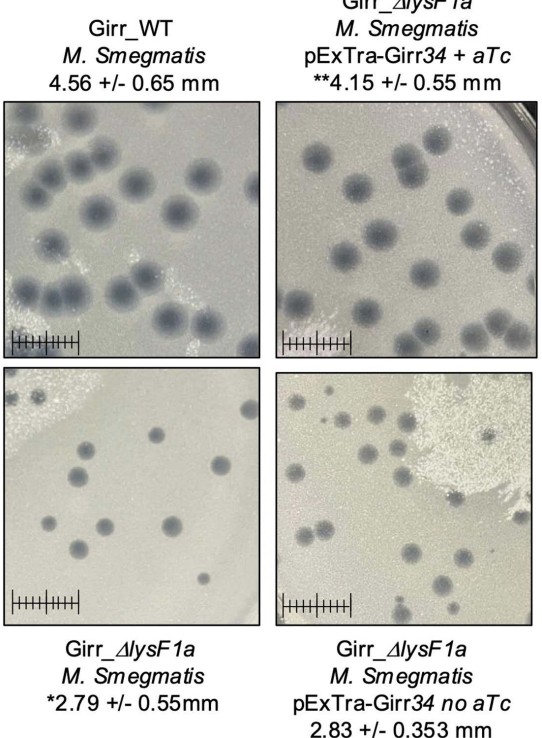

Girr_WT
*M. Smegmatis*
4.56 +/- 0.65 mm

Girr_*ΔlysF1a*
*M. Smegmatis*
pExTra-Girr*34* + aTc
**4.15 +/- 0.55 mm

Girr_*ΔlysF1a*
*M. Smegmatis*
*2.79 +/- 0.55mm

Girr_*ΔlysF1a*
*M. Smegmatis*
pExTra-Girr*34 no aTc*
2.83 +/- 0.353 mm

**Fig 6. Complementation of plaque phenotype in Girr_Δ*lysF1a*.** Plaque assays were completed with *M. smegmatis* using Girr_WT or Girr_Δ*lysF1a* as described in Methods. Plaque assays were completed with *M. smegmatis* complementing strains carrying pExTra-Girr_*34* expression plasmid using Girr_Δ*lysF1a*as described in Methods. Plaques were evaluated following incubation at 37$^{\circ}$C for 48 hrs. Plaques sizes were quantified using ImageJ and the average diameter +/- standard deviation for > 50 plaques is shown. * = statistically different from Girr_WT, *p < 0.0001*. ** = statistically different from Girr_Δ*lysF1a* infected on *M. smegmatis* with pExTra-Girr_*34* with no aTc, *p < 0.0001*. Scale = 1 cm.

ATP release has been used as a measure of cell viability and membrane integrity following phage infection [27,53]. Fig 7C shows the ATP release from the same experiment that is presented in Fig 7A and 7B. As expected, ATP release for Girr_WT, NBJ_WT, Girr_Δ*lysF1a* and NBJ_Δ*lysF1a* generally mirrors the time at which lysis is triggered and the culture OD$_{600}$ begin to decline. ATP levels in the media also peak when the OD$_{600}$ baseline is reached at about 300 minutes (Fig 7C). In contrast, there is delayed but gradual increase in the amount of ATP released from the cultures infected with Girr_Δ*lysF1b* and NBJ_Δ*lysF1b* out to 300 minutes. Although the ATP level does not reach that of Girr_WT or NBJ_WT, the release of ATP confirms that cell integrity has been comprised after infection with the *lysF1b* mutants. To directly assess this contention, aliquots of the cultures were removed at 100- and 260-minutes post infection, serial diluted and plated on 7H10 agar to assess colony growth as detailed in Methods. Both Girr and NBJ are temperate phages, thus, it was expected that the infected cultures would show a small percentage of surviving lysogens as reported in Wise et. al. 2025 [71]. A representative data set with NBJ is shown in S11 Figure 10 in S2 File. Girr showed similar results. Cells infected with NBJ_WT or NBJ_Δ*lysF1b* show >90% reduced colonies 100-minutes post infection when compared to the uninfected culture that was essentially at the same OD$_{600}$. When the cultures were reevaluated at 260-minutes post infection, the colony counts for both NBJ_WT and NBJ_Δ*lysF1b* show a slight 1.4-fold rise due to lysogeny growth, but are still 90% reduced in viable cells compared to the number at the start of the experiment. The lytic phage D29 kills all cells in the culture, validating that the surviving cells are not the result of infection resistant cells or contamination. Thus, this data clearly supports the hypothesis that the lack of a

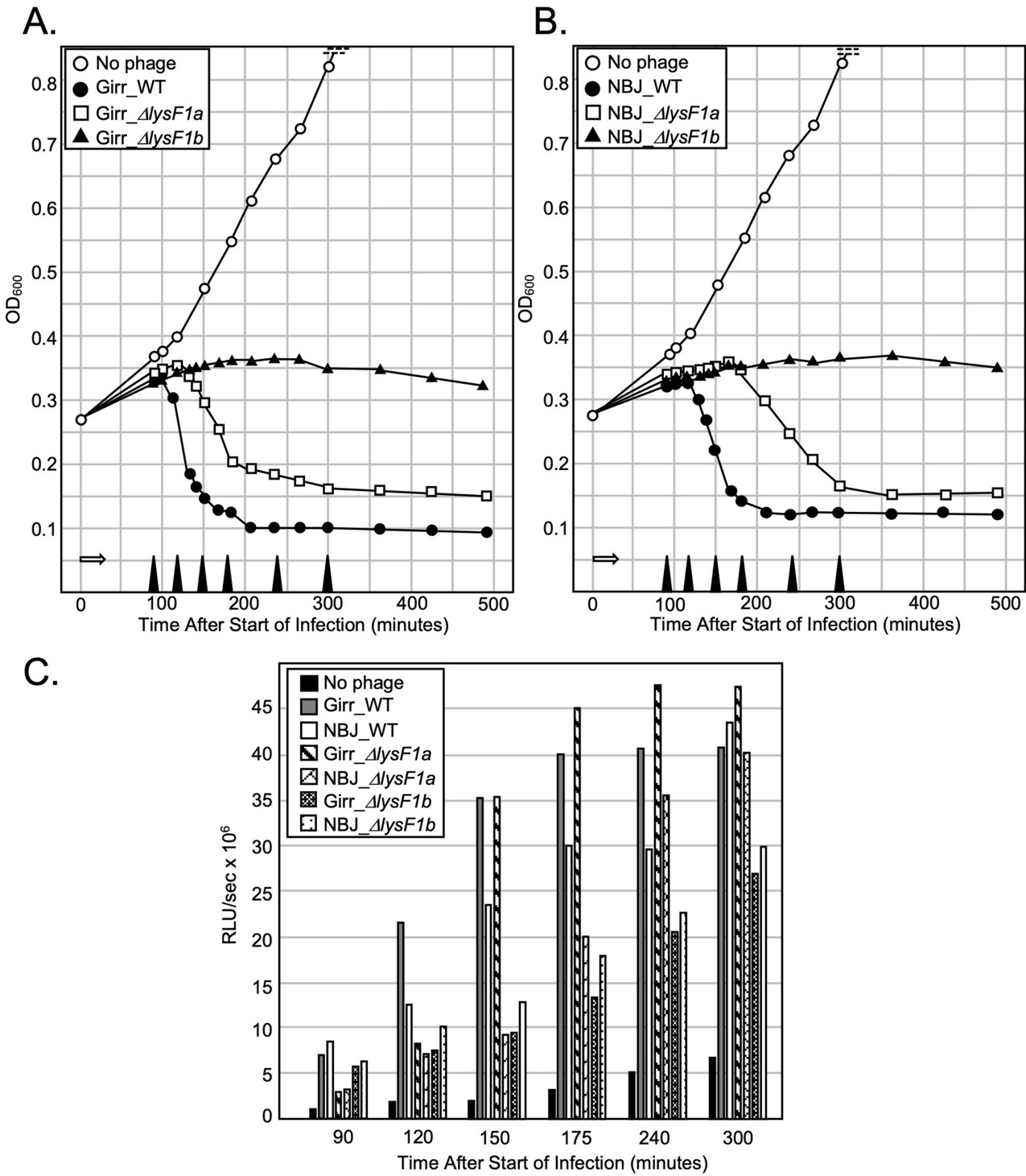

**Fig 7. Liquid lysis assay and ATP release from *M. smegmatis* after infection.** At $T_0$ a log growth culture of *M. smegmatis* at an $OD_{600}$ of ~0.25 was infected with the indicated phage at an MOI of 10:1 and incubated at 37°C for 30 minutes without shaking. Control cultures (no phage) did not receive phage. At $T_{30}$ minutes the culture was constantly shaken a 225 rpm, aliquots removed at the indicated times and bacterial growth evaluated at $OD_{600}$. Open white arrow shows the 30-minute infection time. **A.** Representative results following infection with Girr_WT, Girr_ΔlysF1a and Girr_ΔlysF1b. **B.** Representative results following infection with NBJ_WT, NBJ_ΔlysF1a and NBJ_ΔlysF1b. **C.** Samples of culture were removed from the cultures represented in A and B at the times indicated by the solid black triangle on the x axis. The presence of ATP was evaluated as described in Methods.

defined $OD_{600}$ decline in the Δ*lysF1b* infected cells is due to inefficient lysis and not due to having more viable cells than those infected with Girr_WT.

Collectively, the important finding from this data set is that an F1 phage with only the 1TMD LysF1b protein can trigger the lysis event in the absence of the 2TMD lysF1a protein, but that the lysF1a protein does not support significant bacterial lysis as measured by a drop in $OD_{600}$. The lack of $OD_{600}$ decline is not due to there being a high population of viable cells, although there is a small percentage (<10% of the starting cell population) of surviving lysogens. In fact, the liquid lysis results for Girr_Δ*lysF1b* and NBJ_Δ*lysF1b* is like the lethal *lysin A* deletion in phage Giles where the cultures stopped growing after infection and the $OD_{600}$ gently rose and then plateaued but did not show a sharp triggering decline up through 360-minutes post infection [28]. The similarity in these results suggest that the lysis defects in the Δ*lysF1b* phage may be related to the inability of the endolysins to efficiently cleave the cell wall components because they are either not efficiently released from the infected cell or are exported but cannot be released from the membrane or activated in some manner.

**Energy poisons cause early triggering of F 1Cluster phages with deletion of the *lysF1a* gene but not the *lysF1b* gene in liquid lysis assays**

The results presented so far indicate that the expression of both TMD proteins are required for phage viability, but phages that express only the LysF1b protein can support bacterial lysis while phages that express only LysF1a are viable but severely lysis compromised. The diagnostic assay for holin function is the ability to cause early triggering of an infected culture by the addition of energy poisons (e.g., cyanide or the protonophore dinitrophenol) due to the disruption of the proton motive force (PMF) [11,79]. This type of assay has been utilized in nearly all the pioneering work on holin function of phages that infect *E. coli.* [10,11,19] and both Girr_WT and NBJ_WT can be triggered early if infected cells are treated with CN (S12 Figure 11 in S2 File). To assess the effect of CN on the various mutants, *M. smegmatis* was infected with Girr_WT or Girr_Δ*lysF1a* and the $OD_{600}$ of the culture measured as described in Methods. At 110 minutes after infection, KCN (final concentration of 10mM) was added directly to one culture of Girr_Δ*lysF1a* and to a culture of uninfected cells. Fig 8A shows that Girr_WT triggers at ~110 minutes while Girr_Δ*lysF1a* triggers at 130 minutes just as in Fig 7A. However, when KCN is added to a culture infected with Girr_Δ*lysF1a*, there is a drop in $OD_{600}$ beginning at ~120 minutes that clearly precedes the drop in the untreated Girr_Δ*lysF1a* culture. KCN addition to uninfected cells results in immediate cessation of growth, but not a significant drop in $OD_{600}$. To validate these results, studies were repeated with the NBJ mutants. NBJ_WT triggers later than Girr_WT at ~130 minutes and declines to $OD_{600}$ 0.10 by 300 minutes as expected (Fig 8B). NBJ_Δ*lysF1a* shows delayed triggering just as in Fig 7B and addition of KCN to a culture infected with NBJ_Δ*lysF1a* at 120 minutes results in lysis triggering by 130 minutes with a gradual reduction in $OD_{600}$ that mirrors the NBJ_WT. The ability of Girr_Δ*lysF1a* and NBJ_Δ*lysF1a* to trigger lysis, identifies the 1TMD LysF1b protein as a putative holin and not a releasin, since energy poisons do not trigger lysis prematurely in phage Mu [19].

The next set of studies focused on the phages with deletion of the 1TMD *lysF1b* genes to determine if they could also be triggered by the addition of KCN. Fig 8C shows the results for Girr_Δ*lysF1b*. As previously shown in Fig 7A, *M. smegmatis* infected with Girr_Δ*lysF1b* does not exhibit a defined triggering event as the cells simply cease growing with a slight reduction in $OD_{600}$ out to 480 minutes. Addition of KCN to the Girr_Δ*lysF1b* culture at 180 minutes results in a modest 30% reduction of $OD_{600}$ between 180–240 minutes compared to the Girr_Δ*lysF1b* untreated culture. However, following this initial decline, the level of change in $OD_{600}$ generally mirrors that of the untreated Girr_Δ*lysF1b* culture and the KCN treated *M. smegmatis* control. In contrast, addition of 1% $CHCl_3$ to the Girr_Δ*lysF1b* culture at 180 minutes, results in an immediate precipitous drop in $OD_{600}$ to the level of Girr_WT indicating that the endolysin enzymes are present and active in the cells. The results for NBL_Δ*lysF1b* are shown in Fig 8D and exhibit near identical results to Girr_Δ*lysF1b* in that addition of KCN cause a modest drop during the first hour after addition, and then an $OD_{600}$ decline that mirrors the untreated NBJ_Δ*lysF1b* culture and the KCN treated *M smegmatis*. Addition of $CHCl_3$ results in the same type of sharp decline in $OD_{600}$ to

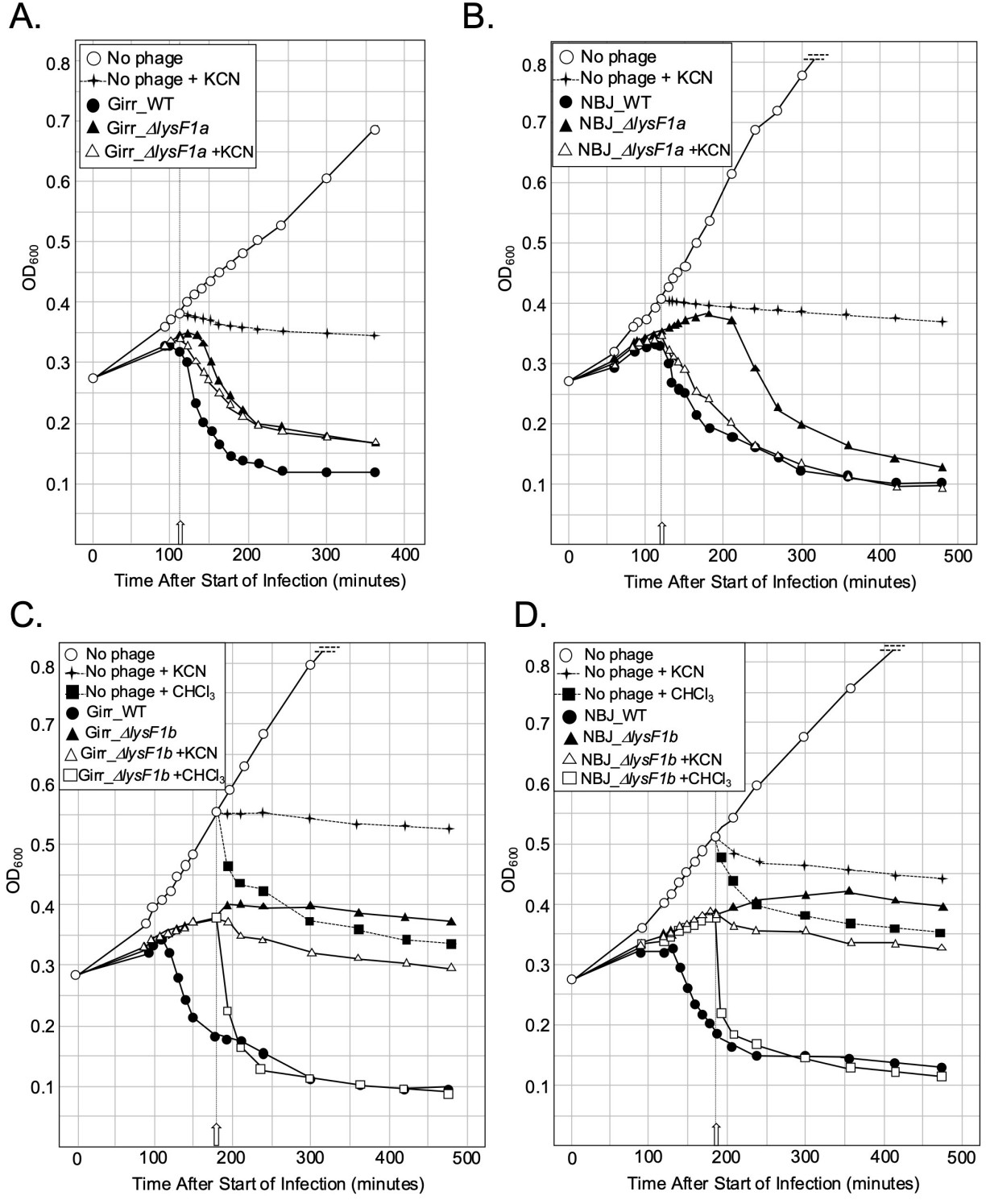

**Fig 8. Energy poisons trigger early lysis by phage expressing LysF1b but not LysF1a.** At $T_0$ a log growth culture of *M. smegmatis* at an $OD_{600}$ of ~0.25 was infected with the indicated phage at an MOI of 10:1 and incubated at 37°C for 30 minutes without shaking. Control cultures (no phage) did not receive phage. At $T_{30}$ minutes the culture was constantly shaken a 225 rpm, aliquots removed at the indicated times and bacterial growth evaluated at $OD_{600}$. **A.** Representative results following infection with Girr_WT, and Girr_ΔlysF1a. KCN (final concentration of 10mM) was added to a culture infected with Girr_ΔlysF1a and an uninfected culture at 110 minutes (open arrow). **B.** Representative results following infection with NBJ_WT, and NBJ_ΔlysF1a.

KCN (final concentration of 10mM) was added to a culture infected with NBJ_ΔlysF1a and uninfected culture at 120 minutes (open arrow). **C.** Representative results following infection with Girr_WT, and Girr_ΔlysF1b. KCN (final concentration of 10mM) or CHCl₃ (final concentration 1%) was added to cultures infected with Girr_ΔlysF1b and uninfected cultures at 180 minutes (open arrow). **D.** Representative results following infection with NBJ_WT, and NBJ_ΔlysF1b. KCN (final concentration of 10mM) or CHCl₃ (final concentration 1%) was added to cultures infected with NBJ_ΔlysF1b and uninfected cultures at 180 minutes (open arrow).

the level of the NBJ_WT baseline. Collectively, these results show that phages expressing only the 2TMD LysF1a protein are not efficiently triggered following disruption of the PMF but that the cells have the requisite enzymes for lysis since lysis occurs immediately after addition of 1% CHCl₃. The inability of the LysF1a proteins to be triggered by KCN is like the $S^{21}71$ or $S^{21}68_{IRS}$ antiholins [13,76,77].

### F1 cluster phage deleted for *lysF1a* have delayed burst timing while phage deleted for *lysF1b* deletion have reduced timing and greatly reduced burst size

The previous results show that infection of *M. smegmatis* with either Girr_ΔlysF1a or NBJ_ΔlysF1a results in a defined triggering (lysis) event that can be prematurely initiated by addition of KCN. However, the results do not establish if phage have been efficiently released from the lysed cells, the timing of the burst event and the burst size. Since it is not possible to utilize lysogens to study phage release in *M. smegmatis*, one-step grow curves were utilized to evaluate the various *lysF1a* and *lysF1b* mutant phages. The experimental set up for these studies is fully described in the methods and representative raw data sets are also shown in S3 Figure 2 and S4 Figure 3 in S2 File. The one-step results from three different experiments utilizing Girr phages are presented in Fig 9. Phage release from Girr_WT shows a lag time of ~180 minutes from the time of infection and then a gradual phage release for 90 minutes before reaching a plateau at ~300 minutes (Fig 9A). The slow release of phage over a 90-minute period is consistent with the liquid lysis assays that showed an ~60-minute timeline of OD₆₀₀ decline after triggering. This is an important observation because there should not be a sharp, immediate lysis of all cells and accompanying phage release since the phage infections are not synchronized but carried out over a 30-minute period and the efficient release of phage from Gram-positive bacteria may also be hampered by the extensive cell wall. Girr_ΔlysF1a shows a delayed lag time by ~20 minutes compared to Girr_WT and a more prolonged period of phage release over 150 minutes before reaching a plateau. The overall burst size for Girr_WT over three experiments was 277 +/- 81 PFU/cell. Girr_ΔlysF1a showed a lower burst of 255 +/- 75 PFU/cell but this was not statistically different from the for Girr_WT. In contrast, Girr_ΔlysF1b shows a very small but defined burst at ~ 210 minutes that is best illustrated when the data is plotted independent of the other phages (Fig 9B). The overall burst size for Girr_ΔlysF1b is only ~29.2 +/- 36 PFU/cell suggesting very poor efficiency of phage release from the infected cells. Overall, this data is consistent with the plaque size and liquid lysis results and supports the hypothesis that *M. smegmatis* integrity is compromised when infected with Girr_ΔlysF1b but the release of phage progeny is severely impacted. In addition, the near wild type burst size observed in cells infected with Girr_ΔlysF1a, further support the holin function of the LysF1b protein.

### Deletion of both *lysF1a* and *lysF1b* is not lethal to phage viability and shows similar phenotypes to the Δ*lysF1b* mutants

Since deletion of either of the individual TMD genes resulted in viable phage, it was of interest to test whether a double deletion of both TMD genes was viable. Both Girr and NBJ were used for these studies and the NBJ data set is shown in Fig 10. BRED was utilized was utilized to delete *lysF1b* from NBJ_ΔlysF1a DNA. Fig 10A and 10B shows the genomic context of the double deletion and the PCR confirmation that both *lysF1a* and *lysF1b* is deleted. The deletion was confirmed by Sanger sequencing of the deleted region and full genome sequencing confirmed that there we no additional nucleotide changes to the genome. A viable double deletion of *lysF1a* and *lysF1b* was also produced in phage Girr that showed identical results to NBJ. Interestingly, while NBJ_ΔlysF1a/ΔlysF1b had dramatically reduced plaque

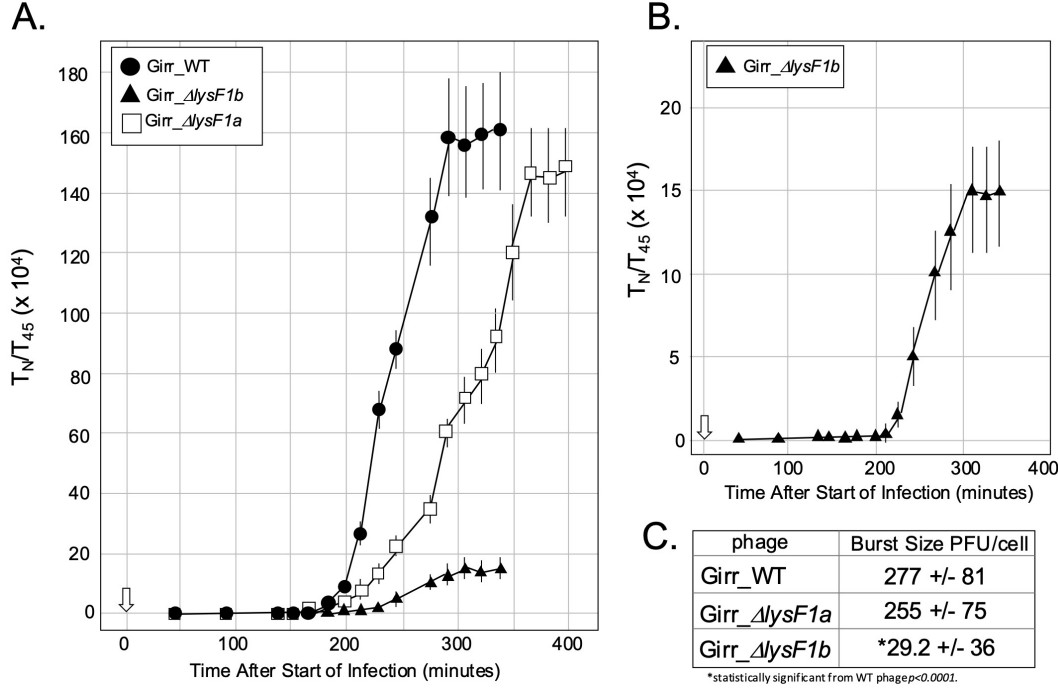

**Fig 9. One-step growth curves of *M. smegmatis* infected with Girr_Δ*lysF1a* and Girr_Δ*lysF1b*. A.** One-step growth curves using Girr_WT, Girr_Δ*lysF1a* and Girr_Δ*lysF1b* were completed as described in Methods. Each time point represents the average and standard deviation from 3 different experiments. **B.** The data for Girr_Δ*lysF1b* from panel A replotted with a different Y axis than used in A. **C.** Burst size was determined as described in Methods. The average bust size +/- standard deviation is from three different experiments and represents the average of all points once the curve reached a plateau.

size compared to the NBJ_WT phage, the plaques were essentially the same size as the single mutation NBJ_Δ*lysF1b* phage (Fig 10C). Liquid lysis data in Fig 10D show that NBJ_WT triggers lysis as expected at ~130–140-minutes while NBJ_Δ*lysF1b* and the NBJ_Δ*lysF1a*/Δ*lysF1b* phage causes reduced culture growth but no defined triggering. Importantly, the addition of KCN was able to prematurely trigger the NBJ_WT but did not trigger the NBJ_Δ*lysF1b* or trigger NBJ_Δ*lysF1a*/Δ*lysF1b.* The finding that the double deletion of *lysF1a* and *lysF1b* is viable and has essentially the same phenotypes as the NBJ_Δ*lysF1b*, support the hypothesis that the LysF1a protein does not contribute to lysis in the absence of LysF1b. These findings also suggest that the F1 phage encoded lysins can be released from the cell independent of LysF1a and LysF1b proteins as suggested for Mycobacteriophage D29 and others [21,30–32],

## Lysis recovery mutants isolated from Girr_Δ*lysF1b* or NBJ_Δ*lysF1b* show wild type plaque size and have point mutations in the *lysF1a* gene

Since both Girr_Δ*lysF1b* and NBJ_Δ*lysF1b* show significant plaque size defects it was possible to screen for lysis recovery mutants (LRM) by assessing changes to the small plaque size following multiple rounds of infection and replication. Fig 11A shows the presence of a large plaque in a Girr_Δ*lysF1b* plaque assay after the second round of harvest and infection.
   When phages were recovered from the large plaque, serial diluted and replated, the wild type plaque size was recovered across the full phage population (Fig 11B). This type of experiment was repeated multiple times with Girr_Δ*lysF1b* and NBJ_Δ*lysF1b* and ten independent Girr_Δ*lysF1b*-LRMs and four independent NBJ_Δ*lysF1b*-LRMs were recovered. All Girr LRM mutants showed a large plaque phenotype compared to the parent Girr_Δ*lysF1b* (S13 Figure 12 in S2 File) and PCR analysis of the *lysF1a-lysF1b* genomic region confirmed that all the LRM mutants contained the expected Δ*lysF1b*

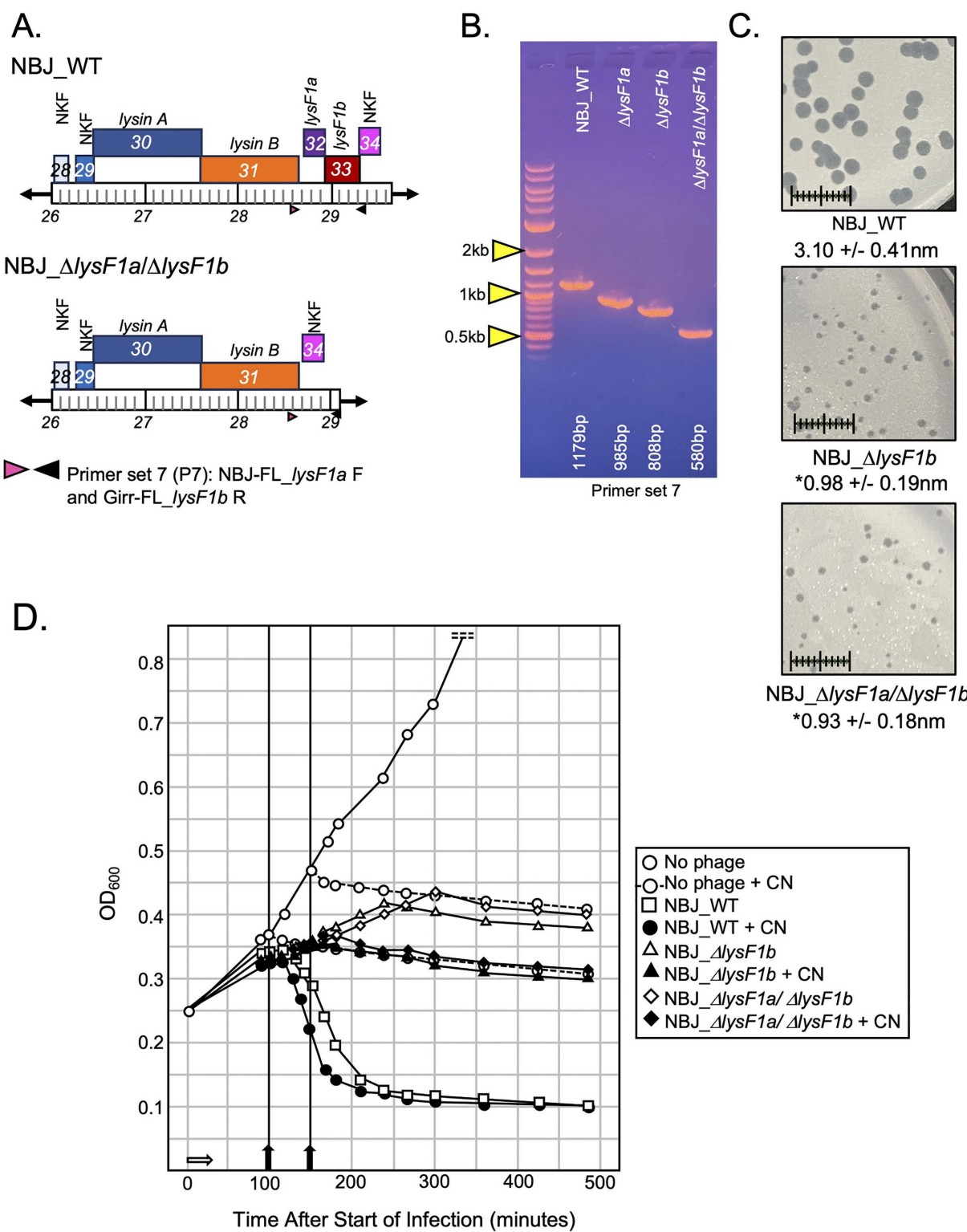

**Fig 10. Deletion of both lysF1a and lysF1b in NBJ. A.** Genomic structure and PCR primer set locations in NBJ_WT and NBJ_ΔlysF1a/ΔlysF1b mutant. **B.** PCR verification of *lysF1a/lysF1b* double deletion. Primer set 7 (P7) uses the forward NBJ-FL_*lysF1a* and reverse NBJ-FL_*lysF1b* and generates a 1179 bp fragment in NBJ_WT, a 985 bp fragment in NBJ_Δ*lysF1a*, 808 bp fragment in NBJ_Δ*lysF1b* and a 580 bp fragment in NBJ_Δ*lysF1a/*

Δ*lysF1b*. **C**. *M. smegmatis* was infected with identical numbers of the indicated phage as detailed in Methods and plaque size determined after 36 hrs. of growth at 37°C. Scale = 1 cm. **D**. Liquid lysis assay. At $T_0$ a log growth culture of *M. smegmatis* at an $OD_{600}$ of ~0.25 was infected with the indicated phage at an MOI of 10:1 and incubated at 37°C for 30 minutes without shaking (indicated by white arrow). Control cultures (no phage) did not receive phage. At $T_{30}$ minutes the culture was constantly shaken a 225 rpm, aliquots removed at the indicated times and bacterial growth evaluated at $OD_{600}$. A separate culture of non-infected cells and NBJ_WT was treated with KCN (final concentration 10 mm) at 100 minutes (black arrow). A separate culture of non-infected cells, NBJ_Δ*lysF1b* and NBJ_Δ*lysF1a*/Δ*lysF1b* was treated with KCN (final concentration 10 mm) at 150 minutes (black arrow).

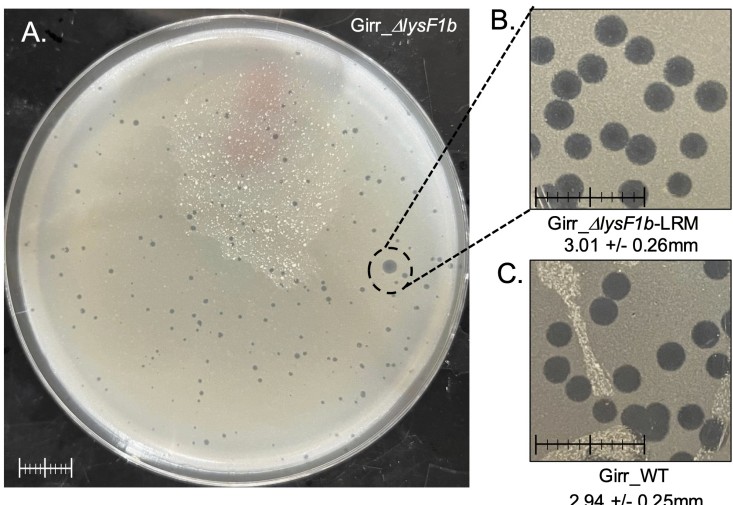

**Fig 11. Isolation of Girr_Δ*lysF1b*-LRM mutants. A**. *M. smegmatis* was infected with GirrΔ*lysF1b*, plate lysates recovered from the plaque assay and *M. smegmatis* infected with the recovered phage. Panel A shows a representative plaque assay following the second round of infection. A large plaque is circled. **B**. Phage were isolated from the large plaque in A, serial diluted and used for plaque assay. Plates were incubated at 37°C for 36 hrs. **C**. Plaque assay from Girr_WT phage. Plates were incubated at 37°C for 36 hrs. Scale bar = 1 cm in all images.

deletion (S13 Figure 12 in S2 File). To determine the location of any nucleotide mutations within the genome of the LRM mutants, DNA was extracted and the genomes were sequenced as detailed in Methods. In all cases, the Girr_Δ*lysF1b*-LRMs and NBJ_Δ*lysF1b*-LRMs contained the expected *lysF1b* deletion but all phage also contained either nucleotide insertions, deletions or substitutions within the *lysF1a* open reading frame that altered the amino acid sequence of the LysF1a protein. A summary of all LRM mutants is provided in Table 3 and a schematic showing the amino acid changes to the different Girr_lysF1a-LRMs is presented in S14 Figure 13 in S2 File.

Six different mutations were identified in the ten Girr_Δ*lysF1b*-LRMs and two in the four NBJ_Δ*lysF1b*-LRMs. The mutations mapped to three different domains of the LysF1a protein: the N-terminal cytoplasmic domain, the first TMD or the beginning of the cytoplasmic C-terminal domain. No mutants were recovered that impacted the second TMD or distal regions of the cytoplasmic C-terminal domain. In the Girr LRM mutants, two different missense mutations were found in TMD1 of LysF1a that result in R19H and A31T while one of the NBL LRM mutants changed the charged cytoplasmic D9 to A. In addition, three G66D mutants were recovered in the Girr LRMs, and three A68E mutants were recovered in the NBJ LRMs. All of these change a small hydrophobic amino acid to a negatively charged amino acid just after the end of the TMD2 at the beginning of the cytoplasmic C-terminal domain. Finally, four mutants generate a nonsense mutation that either truncates the Girr LysF1a protein after G66 or create a frame shift after A63 that results in missense coding and a premature stop after amino acid 73. All LRM mutations result in changes to the LyF1a protein that likely alter its conformation, thus, the findings support that the LysF1a protein can in fact efficiently function in the lysis pathway and that the protein does not exclusively function as an

**Table 3. Lysis Recovery Mutants (LRM) from Δ*lysF1b* phages.**

| Phage | Type of mutation | Genetic location | Mutation Result |
|---|---|---|---|
| Girr_Δ*lysF1b*-LRM1 | point mutation-nucleotide substitution | gene *35 (lysF1a)* | G66D |
| Girr_Δ*lysF1b*-LRM2 | point mutation-nucleotide substitution | gene *35 (lysF1a)* | R19H |
| Girr_Δ*lysF1b*-LRM6 | point mutation- nucleotide deletion | gene *35 (lysF1a)* | missense coding after A63 and premature stop at aa74: A63APRQPRNGFGD.STOP |
| Girr_Δ*lysF1b*-LRM7 | point mutation-nucleotide substitution | gene *35 (lysF1a)* | G66D |
| Girr_Δ*lysF1b*-LRM8 | point mutation- nucleotide insertion | gene *35 (lysF1a)* | stop after G66 |
| Girr_Δ*lysF1b*-LRM11 | point mutation- nucleotide insertion | gene *35 (lysF1a)* | stop after G66 |
| Girr_Δ*lysF1b*-LRM12 | point mutation- nucleotide insertion | gene *35 (lysF1a)* | stop after G66 |
| Girr_Δ*lysF1b*-LRM14 | point mutation-nucleotide substitution | gene *35 (lysF1a)* | G66D |
| Girr_Δ*lysF1b*-LRM15 | point mutation- nucleotide insertion | gene *35 (lysF1a)* | stop after G66 |
| Girr_Δ*lysF1b*-LRM16 | point mutation-nucleotide substitution | gene *35 (lysF1a)* | A31T |
| NBJ_Δ*lysF1b*-LRM1 | point mutation-nucleotide substitution | gene *33 (lysF1a)* | A68E |
| NBJ_Δ*lysF1b*-LRM2 | point mutation-nucleotide substitution | gene *33 (lysF1a)* | A68E |
| NBJ_Δ*lysF1b*-LRM3 | point mutation-nucleotide substitution | gene *33 (lysF1a)* | A68E |
| NBJ_Δ*lysF1b*-LRM7 | point mutation-nucleotide substitution | gene *33 (lysF1a)* | D9A |

antiholin to the LysF1b as suggest earlier. Importantly, plaques were also evaluated for LRM mutants after multiple rounds of plating of the Δ*lysF1a*/Δ*lysF1b* double mutants, but even after six rounds of infection and plating, we failed to identify any changes to plaque sizes, and this suggests that there are no other negative regulators (putative antiholins) of the lysis pathway in these phages.

**Lysis recovery mutants exhibit early lysis and reduced burst size**

Since the LRMs generate a wild type plaque size (Fig 11A; S13 Figure 12 in S2 File), it was pertinent to evaluate the lysis timing in liquid culture and determine lag time and burst size. *M. smegmatis* was infected with Girr_WT, Girr_Δ*lysF1b* and three different Girr_Δ*lysF1b*-LRMs and the $OD_{600}$ of the culture measured as described in Methods. Fig 12A shows that Girr_WT triggers lysis as expected at ~110-minutes while Girr_Δ*lysF1b* causes reduced culture growth but no defined triggering as also shown in Figs 7 and 8. In stark contrast, Girr_Δ*lysF1b*-LRM1, Girr_Δ*lysF1b*-LRM2 and Girr_Δ*lysF1b*-LRM16, all trigger lysis prematurely at ~90 minutes and ultimately reduce the $OD_{600}$ to the same baseline as Girr_WT. When NBJ_WT, NBJ_Δ*lysF1b*, NBJ_Δ*lysF1b*-LRM1 and NBJ_Δ*lysF1b*-LRM7 were tested in the liquid lysis assay, the NBJ-LRMs also trigger lysis prematurely and reduce the $OD_{600}$ to the same level as NBJ_WT (Fig 11B). To validate the early triggering time and assess its impact on burst size, one-step growth curves were completed as described in Methods. Fig 12C shows a representative one-step experiment with Girr_WT and two LRM mutants. Phage release from Girr_Δ*lysF1b*-LRM1 and Girr_Girr_Δ*lysF1b*-LRM2 is detected by 150 minutes and plateaus by 200-minutes, a time point at which Girr_WT has achieved <10% burst size. The two Girr LMRs average burst size is also only 95.5 +/- 12.5 PFU/cell compared to 289.6 +/- 65 PFU/cell for the Girr_WT. The reduced burst size is likely the result of lysis triggering well before all phage progeny have been full assembled and is consistent with reports that have evaluated early triggering in *E. coli* phages [11,12,76]. The current results also indicate that plaque size in our experimental system is not a function of burst size since a > 65% reduction in burst size is still able to generate wild type-sized plaques (Figs 12A; S13 Figure 12 in S2 File). These findings add further support to the hypothesis that the small plaque phenotypes of the various LysF1 mutants is the result of inefficient cell wall lysis that hampers phage release and diffusion.

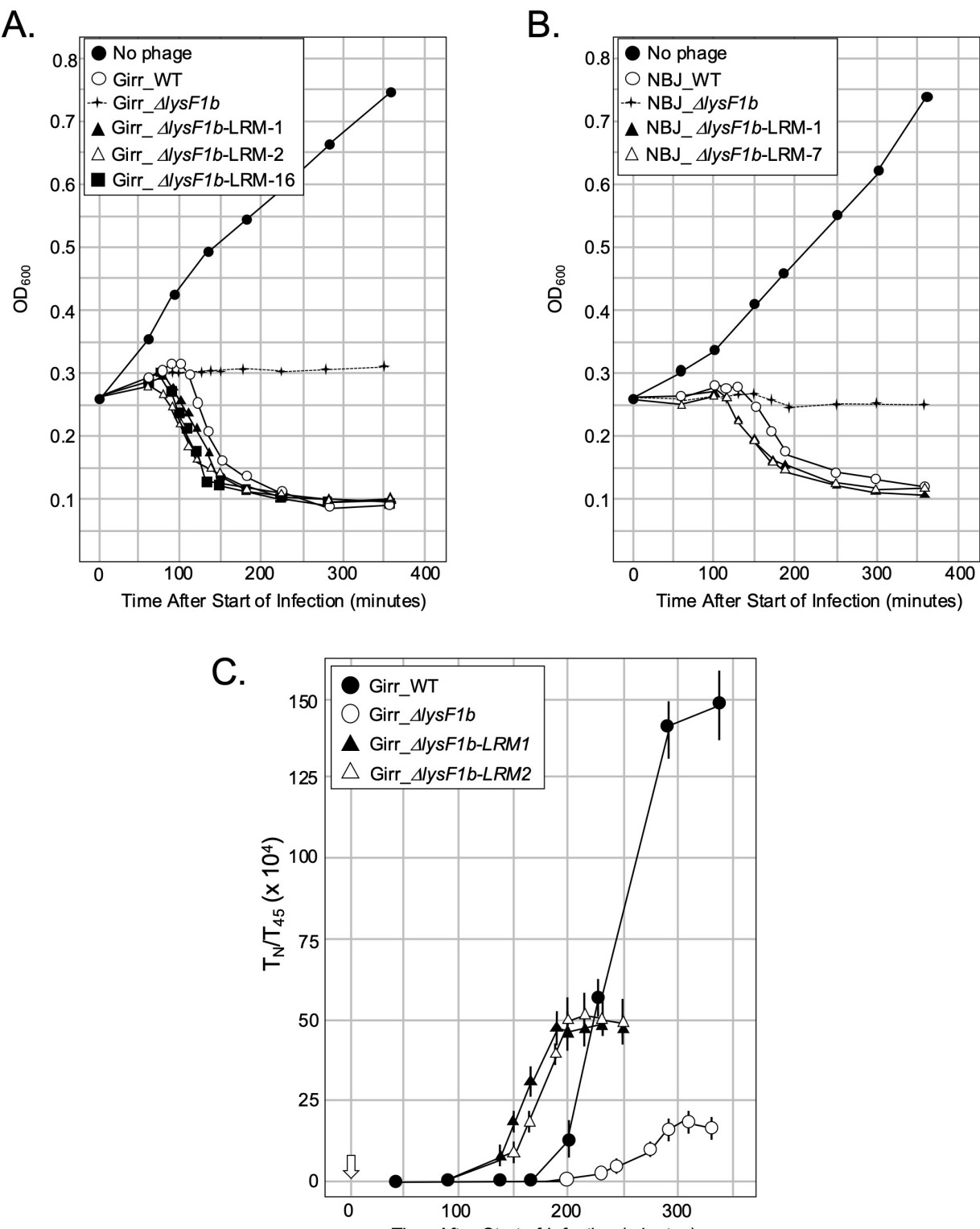

**Fig 12. Liquid lysis and one-step assay of Girr and NBJ LRM mutants.** At $T_0$ a log growth culture of *M. smegmatis* at an $OD_{600}$ of ~0.25 was infected with the indicated phage at MOI 10:1 for 30 minutes at 37°C without shaking. Control cultures (no phage) did not receive phage. At $T_{30}$ minutes the

culture was constantly shaken a 225 rpm and aliquots removed over time. *M. smegmatis* growth was evaluated at OD$_{600}$. **A.** Representative results following infection with Girr_WT and indicated LRM mutants. **B.** Representative results following infection with NBJ_WT and indicated LRM mutants **C.** One-step growth curves using Girr_WT and Girr_Δ*lysF1b*-LRM1 and Girr_Δ*lysF1b*-LRM2 were completed as described in Methods. Each time point represents the average and standard deviation from 3 different experiments.

## Analysis of *lysF1b* phage plaque sizes in *M. smegmatis* lacking the lipoarabinomannan layer of the cell wall

A recent study proposes a 1TMD protein termed LysZ may function as a spanin-like protein to disrupt the integrity of the Gram-positive *Corynebacterium glutamicum* cell wall by interacting with the lipoarabinomannan (LAM) layer [59]. In these studies, deletion of *lysZ* from phages CL31 or Cog results in a small plaque phenotype that is rescued if phage were plated on a *Corynebacterium glutamicum* strain that is defective in the synthesis the LAM. Since both LysZ and LysF1b have a single TMD and deletion of the genes results in small plaques, it was pertinent to determine whether an *M. smegmatis* strain lacking the mtpA enzyme required to synthesize LAM (*M. smegmatis* ΔmtpA) [68], would complement the small plaque phenotype of Girr_Δ*lysF1b*. Wild type *M. smegmatis*, *M. smegmatis* ΔmtpA, and *M. smegmatis* ΔmtpA-comp (ΔmtpA carrying complementing mtpA expression plasmid), were grown to saturation and set to OD$_{600}$ of 2.0 prior to infection. Equal amounts of phage were used to infect and cells and plaques evaluated after incubation for 48 hrs. at 37°C. Fig 13A shows that both wild type Girr and Girr_Δ*lysF1b* exhibit a modest but significant increase in plaque size when infected into the *M. smegmatis* ΔmtpA strain compared to the wild type *M. smegmatis* or the *M. smegmatis* ΔmtpA-comp strain. Girr_WT plaques increased by 1.4-fold from 2.8 mm to 3.9 mm while Girr_Δ*lysF1b* increased by 1.8-fold from 0.93 mm to 1.7 mm (Fig 13B). To account for the possibility that the change is plaque size were caused by reduced cell growth and less *M. smegmatis* ΔmtpA bacteria growing over the courses of the experiment, studies were repeated using high density cultures at 2 x 10$^8$ CFU/infection. This level of bacteria will exacerbate small plaque phenotypes since the increased cells enhance absorption and reduce phage diffusion [28; 80–82]. In these studies, both Girr_WT and Girr_Δ*lysF1b* showed similar plaques size increases as the low-density experiments verifying that *M. smegmatis* lacking a formal LAM layer is generally less restrictive to lysis and phage release. Since both Girr_WT and Girr_Δ*lysF1b* show increased plaque size and the plaques of Girr_Δ*lysF1b* do not reach Girr_WT size, the results indicate that the lack of the LAM layer does not effectively complement the deletion of *lyF1b*. This is consistent with the proposed holin function of the LysF1b and supports the hypothesis that the LysF1b and LysZ proteins have distinct functions in the lysis pathway in their respective bacterial hosts although holin function for LysZ has not been formally evaluated.

## Analysis of additional genes proposed to be involved in the lysis pathway of F1 cluster phages

It is currently unknown how the various endolysins encoded by actinobacteriophages gain access to the peptidoglycan layer following phage infection. Although exogenous expression of the lysin A from some phages is cytotoxic to *M. smegmatis*, there is no evidence of signal sequences or SAR domains in these proteins [21,30–34]. However, *Mycobacterium sp.* appear to have several different secretion systems that do not require a signal sequence [83–86]. Additionally, it has been proposed that phage Ms6 encodes a small chaperone protein (gp23) required for the export of Lysin A because the deletion of the gene results in a small plaque phenotype [50,52]. Since both phage Girr and NBJ encode a homolog to gp23 in the same general genomic area upstream of the Lysin A (gene *29*; Fig 14A), it was of interest to determine if deletion caused a lysis defect in Girr and NBJ. Fig 14D shows that Girr gp29 is 73 amino acids and shares 76.2% identity to Ms6 gp23 while NBJ gp29 shares 76 of 77 amino acids with gp23. BRED was utilized to delete gene *29* from Girr and two independent Girr_Δ*29* phage mutants were isolated (Fig 14B). Girr_Δ*29*–20 and Girr__Δ*29*−25 showed no lysis defect when compared to Girr_WT in the liquid lysis assay (Fig 14C) and showed identical plaque size to Girr_WT (Fig 14E). BRED deletion of NBJ *29* (NBJ_Δ*29*) also showed wild type plaque size (Fig 14E) and no difference in liquid lysis when compared to NBJ_WT.

**A.**

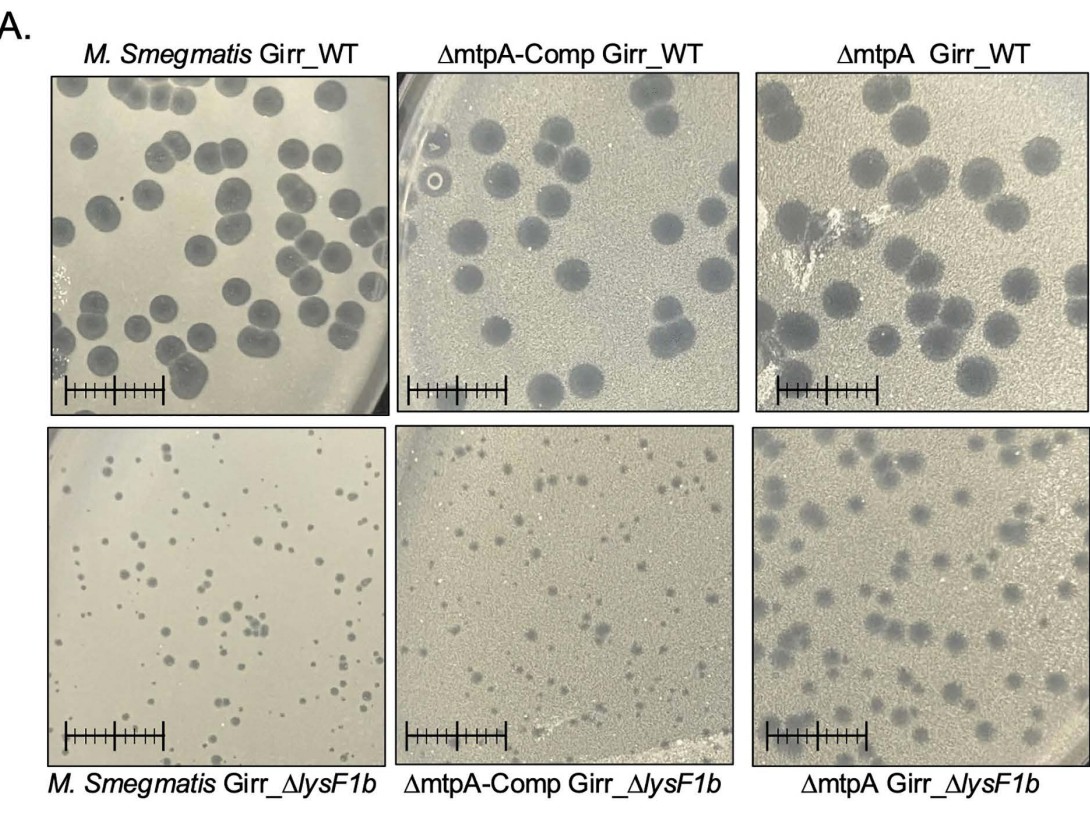

**B.**

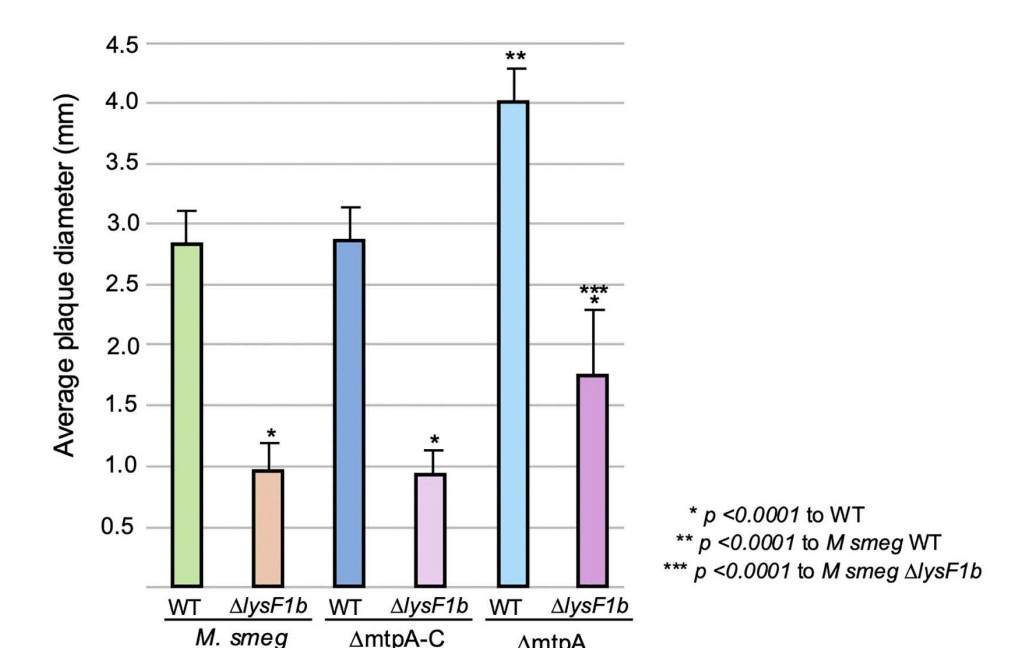

**Fig 13. Analysis of *lysF1b* phage plaque sizes in *M. smegmatis* lacking the lipoarabinomannan layer. A.** Representative plaque assays using Wild type *M. smegmatis*, *M. smegmatis* ΔmtpA-comp or *M. smegmatis* ΔmtpA cultures infected with the same amount of Girr_WT or Girr_ΔlysF1b and incubated for 48 hrs. at 37°C. Ruler = 1 cm. **B.** Plaque sizes were determined as described in Methods. The average plaque diameter +/- standard deviation for 25 plaques is shown.

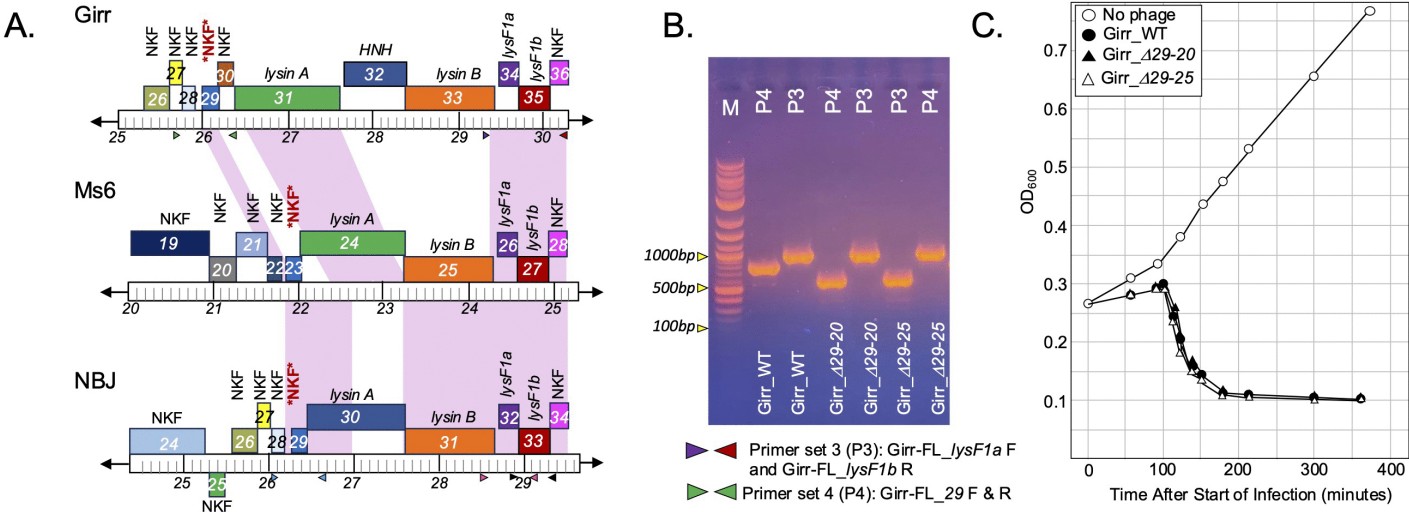

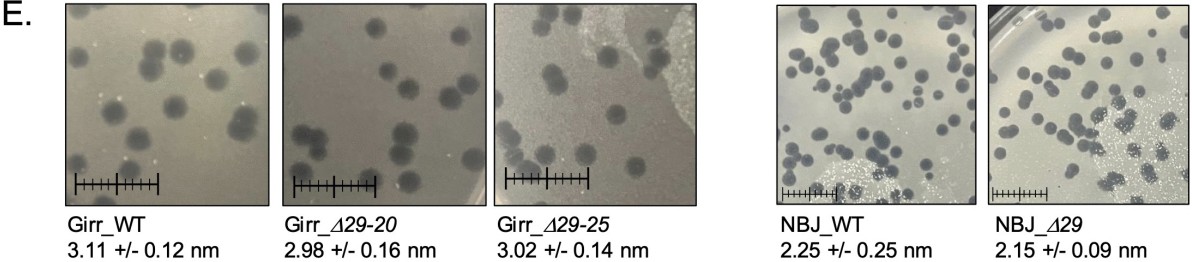

**Fig 14. Analysis of NKF gene *29* in Girr and NBJ. A.** Gene organization of Girr, NBJ and Ms6 lysis cassette region. Predicted protein coding genes are shown as colored boxes. The genomic location is represented by the ruler, and the numbers represent the kilobase from nucleotide 1. Genes above the genome are transcribed in the forward direct and genes below in the reverse direction. Genes with the same color encode proteins that are grouped to the same pham. Pink shading indicated nucleotide identity >90% in the shaded region. *NKF*=Girr_*29*, NBJ_*29* and Ms6_*23*. **B.** PCR analysis of Girr_Δ*29*-20 and Girr_Δ*29*-25. **C.** Liquid lysis assay of the indicated phages was carried out as described in Methods. **D.** Amino acid alignment of Girr gp29, NBJ gp29 and Ms6 gp23 by Clustal Omega. **E.** Plaque assay of Girr_WT, Girr_Δ*29*-20, Girr Δ*29*−25, NBJ_WT, and NBJ_Δ*29*. Ruler=1cm. Plaque sizes were determined as described in Methods. The average plaque diameter +/- standard deviation for 25 plaques is shown.

To explore the possible interaction of the Lysin A and gp29 further, the proteins were modeled using AlphaFold3. The results presented in S15 Figure 14–S17 Figure 16 in S2 File show that all models produced data with ipTM values <0.4 indicating a very low confidence in the interactions of the soluble Lysin A and gp29 proteins. Thus, these results suggest that gp29 from phage Girr or NBJ is not involved in the lysis pathway. A plausible explanation for the lysis defects observed in the Ms6 studies is that the method of deletion of gene *23* somehow impacted the expression of Lysin A (gene *24*) to generate a lysis phenotype. Full genome sequencing data of the Ms6 mutants was not presented or discussed [50,52] and the requirement of a phage encoded chaperone has not been identified in any other phage.

## Implications and future directions

The results presented in this report identify the novel LysF1a and LysF1b proteins as holin-like components required for efficient lysis of *M. smegmatis* by F1 cluster phages. The data support that LysF1a and LysF1b function together to time

the lysis event and disrupt the PMF, however, it appears that LysF1a requires the presence of the LysF1b protein to be active as a putative holin. Holin function is based on several key findings. First, holin function for the LysF1b protein is directly demonstrated by the ability of energy poisons to trigger premature lysis in phages that do not express LysF1a (Fig 8). While it could be argued that the energy poison might trigger lysis by releasing a lysin that was previously exported and tethered to the outer membrane, this hypothesis is not supported by the fact that CN treatment does not trigger lysis in the Δ*lysF1a*/Δ*lysF1b* double mutant. Second, holin function is supported for LysF1b instead of releasin function because there is no defined SAR-endolysins in F1 cluster phages and the energy poisons do not trigger early lysis by releasin in phage Mu [18]. Finally, holin function is supported for LysF1a by the ability to recover LRM mutants in the Δ*lysF1b* background of both Girr and NBJ that map to the LysF1a protein and create phage that trigger lysis early but otherwise are fully lysis competent (Fig 11 and 12).

The 1TMD LysF1b proteins represent a new class of holins and appear to be found predominantly in phages that infect Gram positive hosts. HHpred analysis shows that the LysF1b proteins have a high probability hit to Pfam PF10874 (DUF2746) that contains 906 proteins. Within this Pfam are hits to phages that infect *Mycobacterium sp, Gordonia sp,* and *Streptomyces sp* and hits to *Micromonospora sp, Rothia sp, Saccharothrix sp, Nocardia terpenica, Propionicmonas paludicola, Microbacterium talmoniae, Xylanimonas oleitrophica* and *Actinacidiphilia glaucinigra* where the later hits are likely to proteins that are encoded by prophages within the host bacteria. To assess whether additional phages in the Actinobacteriophage Database at PhagesDB.org express a LysF1b-like homolog [87], phages infecting *M. smegmatis* were bioinformatically evaluated as described in Methods to identify the lysis cassette and associated genes that encode proteins with 1TMD and 2TMD in the same cassette. Of the ~2,600 phages in the database that infect *M. smegmatis*, ~50% (from 22 of the 32 clusters) encode a LysF1b-like protein with a predicted N-out-C-in membrane topology and also have an associated gene encoding a protein with N-in-C-in 2TMD in a defined lysis cassette (S18 Figure 17 and S19 Figure 18 in S2 File). Of the remaining 50% of annotated *M. smegmatis* phages, 30% encode a LysF1b-like homolog but do not have an associated gene that encodes a 2TMD protein, but instead have a gene encoding a protein with 4TMD. Thus, ~80% of the phages that infect *M. smegmatis* have a LysF1b homolog with the predicted N-out-C-in topology. These lysF1a-like proteins are grouped to 20 different phams based on their amino acid identity [88,89]. Representative LysF1a-like proteins from each pham are presented in S20 Figure 19 in S2 File. The proteins differ in size from 92−150 amino acids, have several basic amino acids after the TMD as highlighted in Fig 2 and have between 30%−50% DEKRH charged amino acids in the C-terminal cytoplasmic domain (S20 Figure 19 in S2 File). Multiple sequence alignment indicates minimal amino acid identity between the proteins (S21 Figure 20 in S2 File) and the alignment also failed to identify any domains of high identity. Thus, while ~80% of the phages that infect *M. smegmatis* in the PhagesDB.org database express a LysF1b-like protein, it is unclear whether they all function like the LysF1b proteins from Girr and NBJ.

Since the F1 phages evaluated in this study can clearly perform lysis independent of the LysF1a, and the double deletion of both *lysF1a* and *lysF1b* has the same phenotype as the Δ*lysF1b*, what then is the function of the 2TMD LysF1a proteins in the lysis pathway? As discussed throughout this report, a role of LysF1a as a type II pinholin is not supported by the sequence data (Fig 1B and 1C). In fact, the amino acid sequence data and lack of lysis function in the absence of LysF1b, better supports LysF1a as an pure antiholin like P2 phage LysA [55,56]. However, deletion of the *lysF1a* gene results in a lysis triggering *delay* in both Girr_Δ*lysF1b* and NBJ_Δ*lysF1b* with the later delayed over 60 minutes compared to NBJ_WT (Fig 7 and 8). A lysis delay is not consistent with the function of LysF1a as a pure antiholin to the LysF1b, since it would be expected that deletion of the antiholin would result in *early* host lysis by Girr_Δ*lysF1a* or Girr_Δ*lysF1a* since the LysF1b would not be inhibited from forming active multimers as shown for lambda $S_{105}$ and phage 21 $S^{21}68$ [10–12]. In this context, the key findings are 1) that destabilization of the PMF alone is not sufficient to "activate" LysF1a into a lesion that supports wild type lysis or phage release (Fig 8), and 2) that LRMs can be recovered in the Δ*lysF1b* background that have point mutations LysF1a that result in a fully lysis competent LysF1a protein that happens to trigger premature (i.e., unregulated) lysis (Fig 12). These findings suggest that amino acid changes to the protein may induce

conformational changes to the LysF1a that allow it to fully support bacterial lysis and phage release in the absence of the LysF1b. Thus, an attractive working model is that LysF1a is not an antiholin to LysF1b but simply is not lysis active without LysF1b. Therefore, as expression of both proteins increases following infection, they reach a concentration at which they can formally interact, and this interaction facilitates conformational changes in the LysF1a that allow it to become active in lysis. The two proteins then function together to facilitate the release of endolysins and/or phage particles. Such a model would be like the holin/antiholin system in phage lambda and phage 21 where the phages express both active holins and inactive antiholins and triggering results in an instantaneous conversion of all proteins to active holins [10–12].

Currently, it is unclear how the LysF1a and LysF1b proteins interact and whether both proteins together actively participate in the creation a heteromeric lesion in the inner membrane. Clearly, LysF1b alone can trigger lysis and initiate the process of phage release, but as stated above, lysis timing and the efficiency of phage release are still not optimal. Phage SSP1 appears to have two distinct holins that support lysis when expressed exogenously, but it has not been formally demonstrated that they both form a heteromeric lesion [40]. In addition, the lysis pathway of phage P2 utilizes two distinct proteins, holin Y and antiholin LysA, but LysA protein appears to be fully regulatory to holin Y and is not converted into an active holin like the antiholins $S_{107}$ or $S^{21}71$ [11,57,58]. Thus, the interaction of two distinct holin monomers that have different membrane topologies will require direct study of LysF1a and LysF1b proteins within membranes to resolve the structure. Such work could also directly address whether the LysF1b also has interactions with cell wall components to facilitate phage release as suggested for the LysZ proteins of *Corynebacterium* phages [59]. Whether the other Mycobacteriophages that express different variants of the LysF1a and LysF1b proteins also utilize them in the same way as Girr and NBJ will require additional deletion and complementation studies. The current study therefore provides a framework to evaluate lysis under physiological conditions in Gram-positive hosts. It is exciting to speculate that due to the presence of *lysF1*-like and *lysZ*-like genes in so many phages, it may be that the phages that infect Gram-positive hosts use a similar set of tools to complete the lysis pathway through distinct molecular mechanisms just as observed for phages that infect *E. coli.*

## Limitations

The focus of the current report was to genetically evaluate the function of LysF1a and LysF1b proteins in the physiological context of phage infection in *M. smegmatis*. There are several limitations to the current data set. First, Girr and NBJ are temperate phages and do not kill the entire liquid cultures even at an MOI of 10:1. This is one reason that the culture $OD_{600}$ in our experiments does not drop below 0.10. Although we have shown that the residual surviving bacteria is < 10% of the starting culture and does not impact the ability to assess lysis timing, we have a limited window of analysis of ~6–8 hrs. after infection before the lysogens increase the $OD_{600}$ and begin to interfere with the analysis. Thus, we don't have time points past 480 minutes and it's possible that the various *lysF1b* mutants ultimately lyse the infected cells if allowed to grow for 24 hrs. Second, due to the lysogens present in the experiments, the ability to evaluate the infected bacteria via microscopy (especially electron microscopy) is challenging. Microscopy data is therefore not presented since infected cells and lysogens can't be distinguished prior to lysis and even after lysis the images have lysogens that interfere with the analysis. Third, the LysF1 proteins are integral membrane proteins and AlphaFold3 modeling has reduced confidence for membrane proteins and usually shows low confidence pTM scores of <0.5. Additionally, using AlphaFold3 to assess protein-protein interactions for membrane proteins is also poor with ipTM scores generally <0.5 (low confidence). Finally, the ability to exogenously express LysF1a and LysF1b proteins from plasmids to assess possible interactions and phenotypes in the *M. smegmatis* host is not possible due to the strong cytotoxicity of the LysF1b [70,71].

## Supporting information

**S1 File. S1 Table 1. DNA gBLOCKS and primers.**
(XLSX)

**S2 File. S2 Figure 1 – S21 Figure 20.**
(PDF)

## Acknowledgments

We are grateful to the members of the Science Education Alliance for their invaluable research support and for supplying phage and plasmid reagents, particularly Dainelle Heller, Viknesh Sivanathan, Graham Hatfull, Deborah Jacobs-Sera, Becky Garlena and Dan Russell. We also thank the Pittsburgh Bacteriophage Institute for genome sequencing of the phages described in this study. We want to thank A. McKitterick for helpful discussions regarding LysZ. We want to thank Y. Morita for generously sending the ΔmtpA *M. smegmatis* strains. We thank Welkin Pope for helpful discussions regarding infectious centers.

## Author contributions

**Conceptualization:** Richard S. Pollenz.

**Data curation:** Richard S. Pollenz.

**Formal analysis:** Richard S. Pollenz, Kira Ruiz-Houston, Wynter Dean, Loc Nguyen.

**Funding acquisition:** Richard S. Pollenz, Kira Ruiz-Houston.

**Investigation:** Richard S. Pollenz, Loc Nguyen.

**Methodology:** Richard S. Pollenz, Kira Ruiz-Houston, Wynter Dean, Loc Nguyen.

**Project administration:** Richard S. Pollenz.

**Resources:** Richard S. Pollenz.

**Supervision:** Richard S. Pollenz.

**Validation:** Richard S. Pollenz, Kira Ruiz-Houston, Wynter Dean, Loc Nguyen.

**Visualization:** Richard S. Pollenz, Kira Ruiz-Houston, Wynter Dean, Loc Nguyen.

**Writing – original draft:** Richard S. Pollenz.

**Writing – review & editing:** Richard S. Pollenz, Kira Ruiz-Houston, Wynter Dean, Loc Nguyen.

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
