## [Decision Letter · Decision Letter 0]

18 Feb 2026

Dear Dr. Pollenz,

plosone@plos.org. . . . A letter that responds to each point raised by the academic editor and reviewer(s). You should upload this letter as a separate file labeled 'Response to Reviewers'.A marked-up copy of your manuscript that highlights changes made to the original version. You should upload this as a separate file labeled 'Revised Manuscript with Track Changes'.An unmarked version of your revised paper without tracked changes. You should upload this as a separate file labeled 'Manuscript'.

We look forward to receiving your revised manuscript.

Kind regards,

Hari S. Misra, Ph.D.

Academic Editor

PLOS One

Journal Requirements:

https://journals.plos.org/plosone/s/file?id=wjVg/PLOSOne_formatting_sample_main_body.pdf and and and and https://journals.plos.org/plosone/s/file?id=ba62/PLOSOne_formatting_sample_title_authors_affiliations.pdf

[This project was funded in part by an Undergraduate Grants in Aid for Research from USF Chapter of Sigma Xi to KRH and a USF Internal Proposal Enhancement Grant to RSP.].

4. Thank you for stating the following in your manuscript:

[This project was funded in part by an Undergraduate Grants in Aid for Research from USF Chapter of Sigma Xi to KRH and a USF Internal Proposal Enhancement Grant to RSP.]

[This project was funded in part by an Undergraduate Grants in Aid for Research from USF Chapter of Sigma Xi to KRH and a USF Internal Proposal Enhancement Grant to RSP.]

5. Thank you for uploading your study's underlying data set. Unfortunately, the repository you have noted in your Data Availability statement does not qualify as an acceptable data repository according to PLOS's standards.

At this time, please upload the minimal data set necessary to replicate your study's findings to a stable, public repository (such as figshare or Dryad) and provide us with the relevant URLs, DOIs, or accession numbers that may be used to access these data. For a list of recommended repositories and additional information on PLOS standards for data deposition, please see https://journals.plos.org/plosone/s/recommended-repositories....

6. Please note that your Data Availability Statement is currently missing the repository name. If your manuscript is accepted for publication, you will be asked to provide these details on a very short timeline. We therefore suggest that you provide this information now, though we will not hold up the peer review process if you are unable.

7. We note that you have included the phrase “data not shown” in your manuscript. Unfortunately, this does not meet our data sharing requirements. PLOS does not permit references to inaccessible data. We require that authors provide all relevant data within the paper, Supporting Information files, or in an acceptable, public repository. Please add a citation to support this phrase or upload the data that corresponds with these findings to a stable repository (such as Figshare or Dryad) and provide and URLs, DOIs, or accession numbers that may be used to access these data. Or, if the data are not a core part of the research being presented in your study, we ask that you remove the phrase that refers to these data.

8. PLOS ONE now requires that authors provide the original uncropped and unadjusted images underlying all blot or gel results reported in a submission’s figures or Supporting Information files. This policy and the journal’s other requirements for blot/gel reporting and figure preparation are described in detail at https://journals.plos.org/plosone/s/figures#loc-blot-and-gel-reporting-requirements and https://journals.plos.org/plosone/s/figures#loc-preparing-figures-from-image-files. When you submit your revised manuscript, please ensure that your figures adhere fully to these guidelines and provide the original underlying images for all blot or gel data reported in your submission. See the following link for instructions on providing the original image data: https://journals.plos.org/plosone/s/figures#loc-original-images-for-blots-and-gels.

9. We note that Figures S4, S5, S6, S7, S14, S15, and S16 in your submission contain images which may be copyrighted. All PLOS content is published under the Creative Commons Attribution License (CC BY 4.0), which means that the manuscript, images, and Supporting Information files will be freely available online, and any third party is permitted to access, download, copy, distribute, and use these materials in any way, even commercially, with proper attribution. For more information, see our copyright guidelines: http://journals.plos.org/plosone/s/licenses-and-copyright.

1. You may seek permission from the original copyright holder of Figures S4, S5, S6, S7, S14, S15, and S16 to publish the content specifically under the CC BY 4.0 license.

10. Please include captions for your Supporting Information files at the end of your manuscript, and update any in-text citations to match accordingly. Please see our Supporting Information guidelines for more information: http://journals.plos.org/plosone/s/supporting-information....

Reviewers' comments:

Reviewer's Responses to Questions

**Comments to the Author**

1. Is the manuscript technically sound, and do the data support the conclusions?

Reviewer #1: Yes

Reviewer #2: Yes

Reviewer #3: Yes

2. Has the statistical analysis been performed appropriately and rigorously?

Reviewer #1: Yes

Reviewer #2: Yes

Reviewer #3: No

3. Have the authors made all data underlying the findings in their manuscript fully available?

Reviewer #1: Yes

Reviewer #2: Yes

Reviewer #3: Yes

4. Is the manuscript presented in an intelligible fashion and written in standard English?

Reviewer #1: Yes

Reviewer #2: Yes

Reviewer #3: Yes

Reviewer #1: Comments

The authors presented a manuscript entitled “Genetic Analysis of F1 Cluster Phages that Infect Mycobacterium smegmatis Identifies Two Distinct Holin-Like Proteins that Regulate the Host Lysis Event”. The manuscript is well written, scientifically relevant, and addresses an important topic. This manuscript presents the functions of two small transmembrane holins (LysF1a and LysF1b) encoded by the mycobacteriophages. By recombineering-based gene deletion, plaque phenotype analysis, complementation, and liquid lysis assays, the authors demonstrate that both proteins are holins with different topologies, are individually non-essential but collectively essential for phage viability. The manuscript is well written. Here are a few comments and suggestions.

1. Could the authors determine whether the LysF1a or LysF1b gene alone, when expressed from plasmids, causes growth arrest or membrane permeabilization in M. smegmatis, and do they have synergistic activity when expressed together? Please add topology validation for LysF1a/LysF1b. How common are LysF1a/LysF1b orthologs in Actinobacteriophages? A phylogenetic or topological census would be useful in evaluating the general applicability of the two-holin system.

2. Does the deletion of lysF1 genes affect the timing or extent of Lysin A/B function (e.g., cell wall/membrane), as measured by biochemical or microscopy analyses?

3. Although the paper does reference classical holin literature and mentions Mu releasin and Corynebacterium LysZ, a more general phylogenomic analysis among Actinobacteriophages to place the prevalence of the LysF1a/LysF1b topology would support the argument.

4. More information on lysis in mycobacteriophages, aside from a few previous examples (such as differences in Lysin A/B approaches or the need for envelope modification), would help to contextualize the work.

5. Discuss any technical or methodological limitations and future directions.

Congratulations for the great work.

Reviewer #2: The research article on "Genetic Analysis of F1 Cluster Phages that Infect Mycobacterium smegmatis Identifies

Two Distinct Holin-Like Proteins that Regulate the Host Lysis Event" describes the new dual holin system in F1 cluster phages. The study is well designed, and the data are presented/ discussed well.

1. Line no. 42: The dual-holin system has been previously studied in SPP phages.

2. Line no. 80: Mention the genes involved in the LIN system, rI and rIII.

3. Line no. 82: It will be noteworthy to discuss the T7 phage lysis system. The predicted holin, gp17.5, is found to be non-essential, and no related second holin has been predicted yet.

4. Also mention the class III holin in jumbophage phiKZ; 1TMD, N-in (long) and C-out (short) topology.

5. Table 1: Check for typos. Deletion of gene 29 is mentioned twice.

6. Line no. 404: Usually, AlphaFold3 predictions of membrane proteins are less dependable.

Reviewer #3: Comments to Author:

In the current study, authors have investigated the genomic and functional comparison of Two Distinct Holin-Like Proteins (LysF1a and LysF1b)from Phages Girr and NormanBulbieJr (NBJ) of Mycobacterium smegmatis. They have performed the recombineering for the deletion of both the genes and investigated the plaque morphology in mutant phages. They have also analysed the lysis recovery mutants and did the functional genomics for the same. Authors have done good phage biology experiments, and presented in better way. Further, there are some concern which need to be addressed before final publication.

1. Line 125-155, In introduction section, authors can narrow down the result description, which is already being done in result and conclusion. It apears as repeatation of same information.

2. Advised to merge Fig1 and 2 together.

3. Figure 3 can be moved to supplementary

4. Marker lane in Fig 5B is smeared; it is advised to use another figure with improved marker lane.

5. Figure 9, In graph A, B, curve for no phage seems truncated, and not upto end of kinetics. It is advised to give full curve for better comparison.

6. Figure 10; What is impact of KCN or CHCL3 on wild type phage treated bacterial cell growth.

7. The measurement of CFU count but not absorbance is ideal for viability counts of bacterial cells. Although one can calculate the growth rate from growth kinetics data by using the formula of growth rate (refer this article PMCID: PMC12572221), and can perform the statistical analysis among different group for the statistical significance.

8. It is advised to merge some figure to reduce the number of figure in final manuscript.

.

Reviewer #1: No

Reviewer #2: No

Reviewer #3: **Yes:**Dr. Ganesh Kumar MauryaDr. Ganesh Kumar MauryaDr. Ganesh Kumar MauryaDr. Ganesh Kumar Maurya

---

## [Author Response · Author response to Decision Letter 1]

9 Mar 2026

Reviewer #1: Comments

1. Could the authors determine whether the LysF1a or LysF1b gene alone, when expressed from plasmids, causes growth arrest or membrane permeabilization in M. smegmatis, and do they have synergistic activity when expressed together?

This would be a good experiment but both NBJ and Girr LysF1b is highly lethal when expressed in M. smegmatis from plasmids (see LINES 645-646 and REFS 70 and 71).

Please add topology validation for LysF1a/LysF1b.

We have validated the topologies of all TMD proteins through TOPCONS, DEEP TMHMM and Signal P as stated in the Methods and Referenced to our work on Gordonia phages (REF 46. Pollenz 2022). We also directly show in Figure 2 the topologies of numerous other 1TMD lysis proteins (spanins, holin T and we have now added the 1TMD from phage PhiKZ) and that validate that our analysis can predict the correct topology of these as a positive control to what we show for LysF1a and LysF1b. This level of bioinformatic analysis is consistent with most all previous reports of holin and holin-like TM proteins. The ability to evaluate these via wet lab work in the complex cell wall of Gram-positive bacteria, is beyond the scope of this current report.

How common are LysF1a/LysF1b orthologs in Actinobacteriophages? A phylogenetic or topological census would be useful in evaluating the general applicability of the two-holin system.

In the original paper we included supplemental data regarding the 1TMD LysF1b proteins across the 20 phams from the Actinobacteriophage DB that is the main repository for actinobacteriophages (FIGURES S20 and S21). This was also mentioned in the text (LINES 1145-1152). To add additional detail to this analysis we have now added a new FIGURES S18 and S19 showing schematics of the multi TMD lysis cassettes with 2TMD and 1TMD across 22 of the M. smegmatis phage clusters and report that 80% of the ~2600 annotated phages have a 1TMD homolog to LysF1b. SEE LINES: 1126-1144.

2. Does the deletion of lysF1 genes affect the timing or extent of Lysin A/B function (e.g., cell wall/membrane), as measured by biochemical or microscopy analyses?

We have added a section on LIMITATIONS of our data set and the ability to complete either light or TEM microscopy is not possible due to our phages being temperate and have ~ 10% lysogens in the background. We are currently generating mutants in which the immunity repressor is deleted to create lytic phages of all of our mutants and this will be part of a follow up report.

3. Although the paper does reference classical holin literature and mentions Mu releasin and Corynebacterium LysZ, a more general phylogenomic analysis among Actinobacteriophages to place the prevalence of the LysF1a/LysF1b topology would support the argument.

See the new FIGURES S18 and S19 and expanded narrative LINES 1126-1144. All of the homologs we mention have the same N-out-C-in topology of the LysF1b and the N-in-C-in topology of the LysF1a homologs. We also previously indicated finding the LysF1b homolog in phages that infect Gram-positive bacteria based on HHpred DUF2746 (LINES 1119-1125) and also cite our previous report (REF 46) were we identify several phages with LysF1a and LysF1b homologs in Gordonia phages as well.

4. More information on lysis in mycobacteriophages, aside from a few previous examples (such as differences in Lysin A/B approaches or the need for envelope modification), would help to contextualize the work.

We are unclear exactly what is being asked for here. There is not a deep literature base on mechanistic phage-mediated lysis of Gram-positive bacteria from a genetic and biochemical point of view and we have covered the scope of papers dealing with the few selected phages that have a level of rigor in the analysis. If we are missing other research we are happy to cite it but the great majority of work is bioinformatic or using expression vectors to express phage proteins in E coli. As we state in the paper, the general analysis of the lysis of Gram positive bacteria has been adding lysins directly to growing bacteria. Most data sets measure death, not how the lysins are degrading the cell wall components. The current report deals with the genetic identification and requirement of the phage encoded TMDs, LysF1a and LysF1b that appear to initiate the lysis event. At this time, it is unclear how these proteins facilitate the phage encoded lysins release (if they do) and we discuss this at various points of the report (LINES 192-194, 761-770, 881-883,1061-1113). There are no studies to our knowledge that have directly addressed this complex issue. Our data support that these enzymes can get out of the cells independent of LysF1a and LysF1b since limited lysis still occurs in the double mutant (LINES , 827-830). The various reports from the Jain group in India (REFS 21, 31-33) who work on Lysin A from phage D29 is the most comprehensive analysis of the fact that lysin A can kill M. smeg when expressed from plasmids and we cite these reports. They do not indicate how the lysin from D29 is getting exported when it is expressed from the phage. We are currently not equipped to evaluate membrane dynamics in bacteria undergoing lysis in these very complex Gram-positive bacteria and we strongly feel such work is beyond the scope of this report.

5. Discuss any technical or methodological limitations and future directions.

Congratulations for the great work.

We have added a LIMITATIONS section at the end of the report that address several of the issues mentioned above.

Reviewer #2: The research article on "Genetic Analysis of F1 Cluster Phages that Infect Mycobacterium smegmatis Identifies

1. Line no. 42: The dual-holin system has been previously studied in SPP phages.

The original manuscript referenced SPP1and we have now added a 2nd reference regarding the two possible holins (REF 40) and we discuss the 2 possible holins LINES 1229-1231 although it is still unclear if they both function in making single lesion. We have also removed the sentence from the abstract.

2. Line no. 80: Mention the genes involved in the LIN system, rI and rIII.

We have mentioned these (LINES 93-94, 97).

3. Line no. 82: It will be noteworthy to discuss the T7 phage lysis system. The predicted holin, gp17.5, is found to be non-essential, and no related second holin has been predicted yet.

We have added some additional details about T7 as requested (LINES: 100-103).

4. Also mention the class III holin in jumbophage phiKZ; 1TMD, N-in (long) and C-out (short) topology.

This referenced report was published after we submitted this paper. However, this is a nice suggestion as it provides yet another 1TMD variation to the data set. In this regard we have added its structure to FIGURE 3, referenced the report (REF 78) and mentioned this holin in the narrative (LINES 541-545).

5. Table 1: Check for typos. Deletion of gene 29 is mentioned twice.

Thank you we believe we have rectified this issue.

6. Line no. 404: Usually, AlphaFold3 predictions of membrane proteins are less dependable.

We are aware of these limitations and have a statement regarding this issue to the LIMITATIONS section at the end of the paper and also mentioned prior to the data in LINE 514-515. We still believe it is pertinent to keep this in the report.

Reviewer #3: Comments to Author:

1. Line 125-155, In introduction section, authors can narrow down the result description, which is already being done in result and conclusion. It appears as repetition of same information.

We have deleted several portions of the introduction in question.

2. Advised to merge Fig1 and 2 together.

We have created a hybrid figure 1 as suggested

3. Figure 3 can be moved to supplementary

Since the LysF1b is the key protein in the lysis, we have kept this in the MS as the new Figure 2. We have also added an addition jumbophage holin as it provides additional context to the uniqueness of the LysF1b proteins.

4. Marker lane in Fig 5B is smeared; it is advised to use another figure with improved marker lane.

This figure has been removed.

5. Figure 9, In graph A, B, curve for no phage seems truncated, and not upto end of kinetics. It is advised to give full curve for better comparison.

Any of our data sets that proceed past 360 minutes result in control M. smeg growth that goes beyond OD600 1.0. Thus, we decided to only show the growth up to 0.8. If we plot the entire control bacteria growth out to 8 hrs, the length of the graph will diminish the ability to see the details of the lysis that occurs in infected cultures between OD0.3 and OD0.1. The growth of the bacteria beyond 0.8 also has no impact on the analysis of the triggering time and lysis and simply shows that the bacteria is healthy.

6. Figure 10; What is impact of KCN or CHCL3 on wild type phage treated bacterial cell growth.

Due to the complexity of the liquid death figures and the number of phages we are showing in the data sets, we decided to not include the CN curves for the WT phage as it made analysis of the data very difficult. However, we have included the control data sets with KCN and CHCl3 treatment to the NBJ_WT and Girr_WT on the new FIGURE S12 Figure 11; LINES 780-781). We also present a new FIGURE 10 where we include the KCN treatment for NBJ_WT on this graph.

7. The measurement of CFU count but not absorbance is ideal for viability counts of bacterial cells. Although one can calculate the growth rate from growth kinetics data by using the formula of growth rate (refer this article PMCID: PMC12572221), and can perform the statistical analysis among different group for the statistical significance.

We have added a new FIGURE S11 Figure 10 where we have plated the infected cultures at 100 and 270 minutes to directly assess surviving cells (LINES 742-755). The data conclusively show that the Girr and NBJ phages, regardless of mutation, kill >90 of the cells and we only see a small percent of lysogens that do not impact the data analysis over the time course of the data sets. We also mention the lysogeny of the F1 phages in a new LIMITATIONS section. Thank you for this suggestion.

8. It is advised to merge some figure to reduce the number of figure in final manuscript.

Thank you. We have removed several supplemental figures, but we publish in PLOS due to the ability to provide comprehensive data that show the scope of the entire research project.

---

## [Editor Report · Decision Letter 1]

22 Mar 2026

Genetic Analysis of F1 Cluster Phages that Infect Mycobacterium smegmatis Identifies Two Distinct Holin-Like Proteins that Regulate the Host Lysis Event

PONE-D-25-64520R1

Dear Dr. Pollenz,

We’re pleased to inform you that your manuscript has been judged scientifically suitable for publication and will be formally accepted for publication once it meets all outstanding technical requirements.

Kind regards,

Hari S. Misra, Ph.D.

Academic Editor

PLOS One
---

## [Editor Report · Acceptance letter]

PONE-D-25-64520R1

PLOS One

Dear Dr. Pollenz,

I'm pleased to inform you that your manuscript has been deemed suitable for publication in PLOS One. Congratulations! Your manuscript is now being handed over to our production team.

Kind regards,

on behalf of

Professor Hari S. Misra

Academic Editor

PLOS One